# Site climate more than soil properties and topography shape the natural arbuscular mycorrhizal symbiosis in maize and spore density within rainfed maize (*Zea mays* L.) cropland in the eastern DR Congo

**Adrien Byamungu Ndeko**[1,2,3]*, **Abdala Gamby Diedhiou**[2,3], **Hassna Founoune-Mboup**[3,4], **Géant Basimine Chuma**[1], **Yannick Mugumaarhahama**[1,5], **Diegane Diouf**[2,3], **Saliou Fall**[3,4], **Gustave Nachigera Mushagalusa**[1], **Aboubacry Kane**[2,3]

1 Department of Crop Production, Faculty of Agriculture and Environmental Sciences, Université Evangélique en Afrique (UEA), Bukavu, Democratic Republic of the Congo, 2 Département de Biologie Végétale, Faculté des Sciences et Techniques, Université Cheikh Anta Diop (UCAD), Dakar, Senegal, 3 Laboratoire Commun de Microbiologie (LCM) IRD/ISRA/UCAD, Centre de Recherche de Bel Air, Dakar, Senegal, 4 ISRA_LNRPV, Laboratoire National de Recherches sur les Productions Végétales (LNRPV), Dakar, Senegal, 5 Unit of Applied Biostatistics, Faculty of Agriculture and Environmental Sciences, Université Evangélique en Afrique, Bukavu, South Kivu, Democratic Republic of Congo

* ndeko.byam@gmail.com

## Abstract

Rhizosphere microorganisms, particularly arbuscular mycorrhizal fungi (AMF), play a vital role in enhancing sustainable maize production. However, uncertainty persist regarding the influence of climate variables and soil properties on mycorrhizal colonization (MC) of maize and the abundance of AM fungal spores in the field. This study aimed to explore the environmental factors such as site climate variables, soil physicochemical properties and topography and vegetation variable, affecting the natural MC of maize and the density of AMF spores. The study hypothesizes that natural maize mycorrhizal colonization and AMF spore density vary significantly across different sites and agroecological zones. It further posits that climatic and edaphic variables predominantly explain the observed variation in mycorrhizal parameters. To assess the impact of these factors, a field study was conducted in 32 sites across three territories in the province of South Kivu, namely Kabare, Walungu, and Uvira. Rhizospheric soil and maize roots were collected from different sites. Maize MC varied significantly among sites, with Kabare and Walungu showing high colonization rates (52.1% and 44.7%, respectively) compared to Uvira (26.40%). Meanwhile, spore density was significantly higher in Uvira (1331.7 spores g⁻¹ soil) than in Kabare (518.9 spores g⁻¹ soil) and Walungu (468.58 spores g⁻¹ soil). Correlation analysis indicated that maize MC was influenced by site climate and soil properties. The PLS-SEM model demonstrated that 76.5% ($R^2$) of the total variance in maize root MC was explained by climatic variables and soil chemical properties. Compared to soil chemical properties, climate characteristics had a more pronounced impact on maize MC. Maize MC was inversely correlated with temperature, C and available P content, while being directly and positively correlated with altitude,

**Data Availability Statement:** All of the data underlying the findings of this study are within the paper and its Supporting Information file.

**Funding:** This study forms part of Anova Health Institute's technical support funded by the US President's Emergency Plan for AIDS Relief (PEPFAR) through the United States Agency for International Development (USAID) under Cooperative Agreement number 674-A-12-0. NMZ, ND, CM, AJ, FB and KR received funding through this Cooperative Agreement for their contributions to the research and production of this manuscript. Additionally, this work is supported by IAS – the International AIDS Society with financial support from the Bill & Melinda Gates Foundation (INV-00261 and INV-047567). LSW and AG received funding through this contract for their contributions to the research and production of this manuscript.

**Competing interests:** The authors declare that they have no competing interests.

rainfall, and base saturation rate. Furthermore, 68.5% ($R^2$) of the spore density variability of AMF was explained by climatic variables and soil physical properties. Spore density was inversely correlated with sand and clay content, field capacity, rainfall, and altitude, while being positively correlated with temperature. The results of this study indicate that climatic conditions exert a more pronounced influence on the mycorrhizal colonization of maize and the density of AMF spores than soil characteristics.

## Introduction

The pursuit of improving the maize yields and nutritional qualities has been a global agricultural objectives, especially in the face of climate effects. Maize cultivation in Sub-Saharan Africa "SSA", including the Democratic republic of Congo (DRC), is intricately linked to climate and soil conditions [1]. In this context, predicted drastic changes in climate parameters pose significant risks to food security and agriculture by 2030. Rising temperatures, coupled with increased demand for staple foods, could exacerbate food insecurity if not addressed. It has been demonstrated that a 2°C increase in local temperatures could adversely affect the yields of major cereal crops such as wheat, rice, and maize, resulting in yield losses of up to 25%. Coupled with high demand for staple foods and significant population growth, these disruptions pose a substantial risk of food insecurity in the coming years if no solution is implemented [2]. Moreover, elevated temperatures may decrease soil water levels, leading to drought conditions in most agro-ecosystems [3].

Changes in environmental parameters can directly impact crop development and productivity in various agro-ecosystems [4, 5]. Global variations in climatic and edaphic conditions induce alterations in plant physiology and the emission of root exudates [6]. Specifically, increasing temperatures affect both plant performance and the carbon cycle [7]. In such conditions, symbiotic soil microorganisms, particularly arbuscular mycorrhizal fungi (AMF) are negatively and directly affected, so we can observe a reduction of their diversity and activity. On the other hand, it was reported that AMF develop a mutualistic association with plant roots, play a crucial role in promoting growth, enhancing plant water and mineral nutrition, and protecting plants against diseases and pests by improving tolerance to various biotic and abiotic stress [8]. The role of AMF in improving soil aggregation stability and water retention has also been documented [9, 10]. The effectiveness of mycorrhizal symbiosis in water and nutrient uptake is linked to its capacity to increase the soil volume explored by the root system via the external hyphae of the fungi [11]. Consequently, AMF are recognized as an important component of the soil-plant system and a key element in the sustainable functioning of agroecosystems [12]. Despite their universally recognized beneficial effects, the use of AMF in agriculture remains low in SSA due to inconsistent results in different environments and a lack of experience with this technology. A better understanding of the influence of environmental factors on the establishment of AM symbiosis in maize crops is necessary for the development of sustainable and highly productive agroecosystems [13].

Maize (*Zea mays* L.) holds the position of the primary cereal and the second most significant crop in the Democratic Republic of Congo (DRC), both in terms of production and cultivated area [14]. In 2022, it spanned ~ 2.97 million hectares throughout the entire country. While maize cultivation occurs in all provinces of the DRC, three provinces, namely Maï-Ndombe, Kwilu, and South-Kivu, emerge as major production areas, collectively contributing to over 25% of national production [15]. Despite its importance in Congolese agriculture,

maize yields per hectare have persistently remained low over the past years, with an average recorded yield of ~0.8 t/ha compared to a potential yield of ~7.0 t/ha. This decline is particularly noticeable in the South-Kivu province [15]. Various factors contribute to these low maize yields, including observed changes in environmental parameters in the region over the past 5 years. Locally, these challenges disrupt growing seasons, negative impacting the production of major crops such as maize [16]. In addition to these environmental constraints, cultivated lands are degrading, coupled with a decline in soil fertility, as well as diseases and crop pests, all exacerbated by climate change [17].

Nevertheless, research has shown that maize can establish robust associations with several AMF species, forming effective symbiosis in diverse environments. Under these conditions, maize exhibits increased growth and yield due to the enhanced water and mineral nutrition supplied by AMF [18–20]. Numerous scientific studies have highlighted the positive effects of inoculation with different AMF strains on maize growth and yield [21–23]. Key studies have explored the response mechanisms of genotypes or cultivars to mycorrhizal inoculation, as well as the efficacy CMA strains in various environments. For instance, the study by [24] demonstrated that maize plants inoculated with native mycorrhizal fungi exhibited superior growth compared to non-inoculated plants, emphasizing the high efficacy of local strains over exotic strains. However, the successful and practical application of AMF on maize requires an understanding of the processes involved in establishing mycorrhizal symbiosis in a natural environment, the abundance of indigenous strains, and the factors influencing the establishment of symbiosis in maize concerning the environment [25].

The establishment and efficacy of mycorrhizal symbiosis are directly impacted by environmental parameters. Generally, as per [26], mycorrhizal infections primarily depend on the plant habitat and environmental conditions. Moreover, mycorrhizal colonisation of host plant roots decreases with increased rainfall, as observed by [27]. Similarly [7], found that mycorrhizal colonization is significantly enhanced under water and heat stress conditions. On the contrary [28], demonstrated variations in AMF spore density and root mycorrhization rates among different sites, influenced by altitude, relative humidity, soil pH and available P, K, and Mg content. Regarding temperature effect [29], showed that a temperature increase between 12 and 20°C leads to an increase in the size of the internal and external structures of AMF, thus implying additional carbon costs and high mycorrhization. Additionally [8, 30], found that root colonisation was significantly increased by higher temperatures. On the other hand, sub-optimal temperatures lead to a reduction in root mycorrhization rates and host plant performance by altering the viability of fungal propagules [31]. Previous research suggested that increasing temperature may negatively affect AMF diversity by favoring species that produce spores in higher temperature conditions, ultimately impacting symbiosis functioning [32].

Soil properties like acidity, carbon/nitrogen ratio, and agricultural practices can also influence the abundance of native AMF species associated with maize and the establishment of mycorrhizal symbiosis [33]. High levels of available phosphorus in the soil and high electro conductivity tend to reduce root colonization and AMF spore density [34]. Cropping systems significantly impact AMF efficiency, diversity, and distribution. For instance, studies indicate that crop varieties in low-input farming systems and those adapted to their environments exhibit higher mycorrhizal colonization rates than those in conventional systems [35]. Furthermore, AMF abundance is strongly influenced by land cover and phosphate fertilisation, determining AMF efficiency in agrosystems [36]. Despite, evidence that soil properties may vary according to climatic parameters and altitudinal gradient [37], the combined impact of climatic parameters and soil properties on the abundance of native AMF species and the establishment and functioning of mycorrhizal symbiosis in maize remains unclear in South Kivu

[20]. To date, only a limited number of studies have analysed the influence of climatic parameters and soil properties on the establishment of mycorrhizal symbiosis [19, 38].

In South-Kivu, prior research on AMF has focused on the morphological characterization of AMF species associated with maize and the study of the impact of specific soil properties on AMF diversity [19, 38]. However, to the best of our knowledge, investigations into the natural mycorrhization of maize and the synergistic effects of climatic parameters and soil physico-chemical properties on the establishment of mycorrhizal symbiosis have not been undertaken. Existing evidence indicates that AMF species are ecologically and functionally distinct and respond differently to variations in environmental factors, potentially influencing mycorrhization process [39]. A comprehensive understanding of the natural mycorrhization of maize and the sporulation of AMF strains in diverse South Kivu environments is crucial for determining different sporulation profiles and identifying suitable areas for the development of mycorrhizal symbiosis in maize plants, considering environmental factors. Consequently, the objectives of this study are to (i) present initial insights into the natural mycorrhization of maize in South-Kivu and (ii) analyze the combined effect of different climatic factors and soil physico-chemical properties on the natural mycorrhization of maize and spore density in the maize rhizosphere. We hypothesized that natural maize mycorrhization and AMF spore density will vary significantly between the selected sites and across the agroecological zones of South Kivu. Additionally, we propose that the climatic and edaphic variables more than topographic and vegetation variables of the sites will account for the variability observed in natural maize mycorrhization and AMF spore density in South Kivu.

## Materials and methods

### Field site description

The study was conducted in the South-Kivu province, located in the eastern Democratic Republic of Congo (DRC). Specifically, we focused on three territories mainly Kabare, Walungu, and Uvira. These regions are situated within three distinct agroecological zones (AEZ) where maize serves as the predominant staple food crop, primarily cultivated under natural rainfall conditions. These areas align along an altitudinal gradient, resulting in significant diversity n environmental and geographical characteristics. Nevertheless, similarities between Kabare and Walungu are evident, as depicted in the comparative table [40]. The first and second zones, namely Walungu and Kabare, are located at high and mid altitudes, while the third zone, Uvira, is located at a lower altitude. Kabare territory lies between 28˚45' - 28˚55' East longitude and 2˚30' - 2˚50' South latitude, ranging from 1,460 to 3,000 meters above sea level. Covering an area of approximately 1960 square kilometers, Kabare experiences a moderately hot and humid tropical climate, with clay loam soil that is fertile but susceptible to water erosion and other forms of degradation. The average annual rainfall in this region is 1601 ± 154 mm, and the monthly temperature averages around 19.67 ± 2.3˚C [41].

The Walungu territory spans between 28˚24.6' and 28˚45.5' East longitude and 2˚41.5' and 3˚ South latitude, with altitudes ranging from 900 meters (at Kamanyola) to 3000 meters above sea level. This region receives an average annual rainfall of 1,600 mm, with monthly maximum temperatures reaching 25˚C and average minimum temperatures of 18˚C. Uvira, contrastingly, exhibits a semi-arid climate classification according to the Köppen-Geiger system. It is situated between 29 and 29˚30' East longitude, spanning 3˚20' to 4˚20' South latitude and ranging in altitude from 800 to 900 meters above sea level. This area receives an average annual rainfall of 900-1000mm, with average temperatures varying between 30.5 to 32.5˚C and 14.5 to 17˚C. The climatic patterns in these three territories delineate two seasons. The

prolonged rainy season prevails from September to June, followed by a brief dry season lasting three months, occurring from July to August.

In this area, 32 sites of maize (*Zea mays* L.) were randomly selected, based on their location as described in the methodological framework (S1 Fig): latitude, longitude and elevation, and their bioclimatic characteristics. The different sites were grouped according to the altitude gradient, a criterion used to describe the agro-ecological zones (Lowland and Highland) in South Kivu province [42]. Consequently, 11 sites were located at low altitude (Uvira) in lowland, 13 sites were located at medium altitude (Kabare) and eight sites were located at high altitude (Walungu) in highland. For each site, environmental, topographic and edaphic parameters were generated, and recorded (Table 1) during the period of the investigation (two months

**Table 1. Description of climatic, soil properties, and mycorrhizal parameters considered in the three agroecological zones (AEZ) in the South-Kivu province.**

| Latent variables | Measured variables | Abbreviations |
|---|---|---|
| Predictor variables | | |
| Site climate parameters | Elevation range (m a.s.l.) | EL |
| | Average annual rainfall (mm) | AR |
| | Average monthly temperature (˚C) | Tmean |
| | Average monthly minimum temperature (˚C) | Tmin |
| | Average monthly maximum temperature(˚C) | Tmax |
| | Average air relative humidity (%) | ARH |
| | Wind speed (m/s) | Wind |
| | Solar radiation (W/m$^2$/s) | SRadiation |
| | Vapor pressure (Pa) | VaporP |
| Soil chemical properties | Soil pH-H$_2$O | pH |
| | Soil organic carbon content [%] | C |
| | Soil organic nitrogen [mg g$^{-1}$] | N |
| | Soil available phosphorus [ppm] | P |
| | Soil Potassium content (mg kg$^{-1}$) | K |
| | Soil Calcium content (mg kg$^{-1}$) | Ca |
| | Cations exchange capacity (meq/100g) | CEC |
| | Soil Sodium content (mg kg$^{-1}$) | Na |
| | Soil Magnesium content (mg kg$^{-1}$) | Mg |
| | Base saturation rate | TSB |
| Soil physical properties | Soil clay content [%] | SC |
| | Soil sand content [%] | SS |
| | Soil bulk density | Density |
| | Soil Field Capacity | Fieldcap |
| | Soil water content | WC |
| Topographic parameters | Altitude (in m) | Alt |
| | Normalized Difference Vegetation Index | NDVI' |
| | Compound Topographic Index | CTI |
| | Slope (%) | Slope |
| | Curvature | Curvature |
| | Slope Aspect | Aspect P |
| Response variables | | |
| Mycorrhizal status | AMF root colonization (Mycorrhizal Frequency %) | Freq |
| | Mycorrhizal colonization rate (Mycorrhizal intensity %) | Itens |
| | AM Spore density (number of spore/100 gr soil) | Sp_dens |

after the beginning of the crop season, in October 2020 in highland and December 2020 in lowland).

## Soil and root sampling

At each site, three maize fields were selected along a diagonal transect, ensuring a minimum distance of 500 meters between them. Sampling took place during the rainy season from October 2020 to December 2020 when the maize plants reached three month growth across all sites. Within each maize field, samples were collected following a diagonal sampling pattern, each neighbouring sample separated by 10 m. Each sample was composed of five plants, so fifteen plants per site were collected. The maize plants were delicately uprooted to preserve the integrity of the root system. A circle approximately 15 cm in diameter was traced around each plant, and the soil was dug to extract the entire root system.Collected roots, along with the surrounding rhizosphere soils (0-20-25 cm), were combined to create three composite samples. In total, ninety six soil and root samples were collected across the thirty two sites. Roots were separated to isolate the rhizosphere soils, washed with tap water, preserved in 75% ethanol, and stored at 4°C for subsequent evaluation of arbuscular mycorrhizal fungi (AMF) colonization. Soil samples were also stored at 4°C for spore extraction, while another portion was kept at room temperature for analyzing the soil's physicochemical properties.

## Assessment of AMF root colonization

In order to quantify the level of root colonization of maize by indigenous AMF strains, the roots samples underwent initial rinsing with tap water followed by distilled water to eliminate any accumulated alcohol from the storage process. Fresh roots were then assessed for AMF colonization using the [43] method, as outlined by [8]. The procedure involved root clearance in 10% KOH solution, subsequent rinsing in distilled water, acidification in 1% HCl, and staining with 0.05% trypan blue, followed by incubation in a 90°C water bath. Trypan blue solution used was prepared in lactophenol. The total mycorrhization percentage was determined using the method outlined by. Root fragments from the 96 field samples collected across all study sites were mounted on microscope slides and observed through an A-Plan $40 \times /0.65$ objective. The mycorrhizal root length percentage was assessed by examining 1 cm root fragments following the procedure outlined by [44]. Each site underwent 300 observations (100 root fragments per sample $\times$ 3 replicates) in order to achieve a more accurate determination of the maize mycorrhizal colonization rate (MC%). The percentage of mycorrhization (frequency (F) and intensity (I)) was calculated using the formula below;

$$F(\%) = \frac{\text{Number of mycorrhized root fragments}}{\text{Total number of observed root fragments}} \times 100 \tag{1}$$

$$I(\%) = \frac{95\,n5 + 70\,n4 + 30\,n3 + 5\,n2 + n1}{\text{Total number of observed root fragments}} \tag{2}$$

Where n1, 2, 4, 4, 5 designated the number of fragments scored 1 ($<$1%), 2 ($<$10%), 3 ($<$50%), 4 ($>$50%), and 5 ($>$90%).

## AMF-spore isolation and spore density determination

AM Spores were isolated from the 96 soil samples (with 32 sites and 3 replicates each). The soil underwent thorough homogenization, and any larger stones or roots were sieved out using a 2 mm sieve. From the sieved soil, only 100 g was carefully selected for spore isolation. This process

involved a specialized method that combined wet sieving, decanting (using sieves of sizes 500, 300, 100, and 40 μm), and sucrose centrifugation. Centrifugation was performed with two sucrose solutions, 20% and 60%, to isolate the AMF spores. The approach was adapted from the methods outlined by [45, 46]. The material obtained was collected from the 40 μm sieve and transferred to a plastic Petri dish. The dish was marked with a 5 mm square grid, enabling the separation of stereomicroscope fields for spore counting. Spore counting was directly performed using a stereomicroscope at 40× magnification. Only intact and visibly healthy spores were tallied, and sporocarps and spore clusters were treated as a single spore unit.

## Soil analysis

The soil samples were first dried and sieved through a 2 mm sieve before undergoing analysis. Soil pH was measured in a soil-water suspension at a 1:1 ratio [19]. Total nitrogen was determined using the Kjeldahl method, using sulfuric acid and selenium, followed by distillation and titration of the produced ammonia, while organic carbon and organic matter (organic matter = 1.725 * C%) were analyzed using the Walkley-Black method [47]. Exchangeable cations were measured using a flame photometer and a spectrophotometer in ammonium acetate solution [48]. Soil available phosphorus content was determined using the Olsen method (extraction with sodium bicarbonate with detection using molybdenum blue) [49]. Additionally, soil variables from the ISRIC database (https://www.isric.org/) were utilized, including bulk density (in tons per cubic meter), clay, silt, and sand contents, cation exchange capacity (CEC), as well as soil water retention capacities (Field Capacity).

## Environmental parameters description and acquisition

The detailed list of environmental and mycorrhizal variables utilized in this study is provided in **Table 1**. Environmental variables in each site were extracted from the local station and other on Worldclim database over the sample collection period. The topographical variables included slope inclination (in %), elevation (in m), slope aspect, Topographic Witness Index (TWI) and plan curvature, these variables were derived from ALOSPALSAR digital elevation model (DEM) images of 12.5 m spatial resolution. Value extraction were made using spatial analysis in ArcGIS 10.7.1 Esrti-TM. Climatic variables included the annual mean, minimum and maximum of temperature (in ˚C), precipitation (mm/year), solar radiation (KJ m$^{-2}$ day$^{-1}$), and wind speed (m/s). Regarding land use and land cover (LULC) variables, they were extracted from the south-Kivu raster image produced by the "Observatoire des Forets et Paysages Montagneux". Two values were extracted mainly the land use and land cover and the NDVI index. In fact, the different LULC are detected by remote sensing using supervised classification algorithms. ArcGIS 10.7 Esri-TM.

## Data treatment and analysis

**Descriptive statistic and analysis of variance (ANOVA).** The means and standard deviations of mycorrhizal variables are succinctly presented in **Table 2 and Figs 2 and 3**, while environmental variables are explicitly outlined in **S1 Table** and summarized on **Fig 4**. In order to assess the distribution of quantitative data, a normality test utilizing Shapiro and Bartlett tests were meticulously conducted using XLSTAT version 2020.1.3. Subsequently, a one-way analysis of variance (ANOVA) was applied to assess the variation in both mycorrhiza and environmental variation across the diverse study sites and territories. Turkey Honestly Significant Different (HSD) test was used to compare means at 5% threshold. This approach allowed us to determine the variation in maize root colonization and spore density across different sites. Pearson correlation was applied to assess the relationship between explanatory variables and

**Table 2. The mycorrhizal colonization of *Zea mays L* roots, measure in terms of mycorrhization frequency and mycorrhizal intensity, along with spore density, assessed across thirty-two sampling sites.**

| Territories | Sites | Mycorrhizal frequency (Mean±SD) | Mycorrhizal intensity (Mean±SD) | Spores Density/100g Soil |
|---|---|---|---|---|
| Kabare | Bugobe cifuma | 61.9±4.2ab | 18.1±2.8ab | 321±19bcd |
| | Bugobe kahave | 61.1±4.2ab | 13.7±1.8bc | 62±15k |
| | Bushumba | 60.6±5.3ab | 11.5±0.8bcd | 263 ± 4 defg |
| | Bushwira CE | 59.2±2.5ab | 13.1±1.4c | 234±21defgh |
| | Bushwira Cit | 48.8±5.3bcde | 11.1±0.9cde | 309±7cdef |
| | Cirunga Kar | 68.3±2.3a | 22.4±3.8a | 212±15efghij |
| | Cirunga Mul | 41.7±5.8cdef | 10.7±0.7cdefg | 210±7efghij |
| | Katana | 48.3±7.6bcde | 10.8±1.6cdef | 122±8ijk |
| | Kavumu | 37.7±2.5defgh | 12.9±1.8c | 59±14k |
| | Luhihi | 48.7±6.4bcde | 11.3±0.4cd | 130±11hijk |
| | Miti | 58.1±3.5ab | 10.4±0.9cdefgh | 47±42k |
| | Mudaka | 41.3±3.5cdef | 10.3±0.9cdefghi | 71±19k |
| | Tchibati | 41.7±3.5cdef | 9.6±0.5cdefghi | 110±9jk |
| Uvira | Bwegera | 22.7±3.3hij | 6.6±0.7defghi | 431± 23a |
| | Kabunambo | 22.8±6.8ghij | 5.5±0.3i | 381±47abc |
| | Katogota | 34.1±4.1fghij | 5.8±0.06ghi | 417±24ab |
| | Kigurwe | 30.1±8.1fghij | 6.3±0.15efghi | 379±10abc |
| | Kiliba | 25.7±4.3fghij | 6.3±0.8efghi | 291±13cdef |
| | Lubarika | 22.8±5.2hij | 6.9±0.2defghi | 395±26abc |
| | Luvungi | 36.3±4.1efghi | 6.7±1.1defghi | 310±33cde |
| | Ndunda | 18.8±2.2j | 5.9±0.3fghi | 258±43defg |
| | Rusabagi | 21.2±5.7ij | 5.7±0.7hi | 215±16efghij |
| | Sange | 22.1±5.9hij | 6.2±0.2fghi | 268±7defg |
| | Sasira | 33.8±4.5efghij | 6.3±0.3efghi | 271±5defg |
| Walungu | Burhale | 38.9±3.8defg | 13.6±1.9bc | 159±8ghijk |
| | WC | 40.1±5.3cdef | 10.2±0.9cdefghi | 74±6k |
| | Butuza | 53.2±8.4abcd | 19.1±4.2a | 421±65ab |
| | Kamanyola | 28.3±7.4fghij | 10±0.4cdefghi | 225±33defghi |
| | Luciga | 59.5±3.8ab | 12.7±0.5c | 67±27k |
| | Lurhala | 54.9±6.1abc | 18.9±3.1a | 126±10ijk |
| | Mugogo | 36.7±6.1afghi | 10.6±0.9cdefgh | 81±17k |
| | Mushinga | 46.2±5.1bcde | 11.4±1.2cd | 92±10k |
| *p-value* | *Territory* | <0.0001 *** | <0.0001 *** | <0,0001 |
| | *Site* | <0.0001 *** | <0.0001 *** | <0,0001 |
| | *HSD* | 16.06 | 4.8 | 34 |

ANOVA: analysis of variance, ns: non-significant differences (p≥0.05)

**: $p<0.05$ (significant)

***: $p<0.01$ (highly significant), and

****: $p<0.0001$ (very high significant). Values of each colon with different letters are significantly different according to Tukey HSD test at 5% threshold.

the explained variables (specifically the mycorrhization rate of maize and the spore density). This analysis was performed in R software version 4.3.2.

**Multiple factor analysis (MFA).** Multiple factor analysis (MFA) was applied to the variable groups to establish the linear relationships between them. All the study variables, including soil physico-chemical properties, climatic characteristics, topographic and mycorrhization parameters (mycorrhizal colonization and spore density), were considered. Before conducting

the MFA, we initially assessed the appropriateness of the analysis for the groups of variables under study. High correlation or independence between variables might suggest that MFA is not suitable for the dataset [50, 51]. To determine this, we applied Bartlett's Sphericity Test and the Kaiser-Maier-Olkin (KMO) test. A KMO index value exceeding 0.6, along with a *p*-value below 0.05 for Bartlett's sphericity test, indicates the suitability. The selected variables were then utilized to construct principal components through MFA [52, 53]. Following Kaiser's "absolute" criterion, all factors (components) with eigenvalues greater than 1 were retained [54]. The MFA procedure was implemented using the FactoMineR and Factoextra package in R software version 4.3.2.

**Partial Least Squares Structural Equation Modeling (PLS-SEM).** To evaluate the significance and assess the importance of the effect of environmental factors on mycorrhizal colonization and spore density, we employed the Partial Least Squares Structural Equation Model (PLS-EM) on all variables in this study. PLS-SEM is a widely used method for estimating path models with latent variables and their relationships. It combines principal components analysis with multiple regression-based path analysis [55, 56]. One of its key advantages is its flexibility, as it does not impose restrictions on data normality or sample size, making it effective even with small samples. Additionally, it allows the use of unmeasured variables as latent variables estimated from measured variables (manifest variables) in the model [57]. In this study, predictor variables included physicochemical properties, topographic, and climatic variables were used as predictor variables, while mycorrhizal colonization and spore density per site were considered separately as response variables. First we defined the different latent variables and their respective scores following the instructions described by [58]. We constructed a heat map of the studied variables to identify those that would be relevant before executing the SEM model (**S2 Fig**). Thus, each latent variable consisted of a group of measured variables or manifest variables. In this study, the SEM model was applied firstly to test the effect of predictor parameters on AMF root colonization, and secondly to test the effect of these same factors on AMF spore density.

The validity of the selected model was evaluated using three criteria, including the average variance extracted (AVE), composite reliability (CR), and the $R^2$ value respectively. AVE and CR values exceeding 0.5 and 0.7, respectively, are considered acceptable [59], and in this study, the outputs of the PLS-SEM model adhered to these standard criteria, with AVE>0.5 and CR>0.7 (**Figs 7 and 8**). These analyses were conducted using the PLS-SEM model from SmartPLS software version 4.1.0.2 [60]. We used bootstrap standard errors to determine the statistical significance of the original indicator weights, loadings, and path coefficients [61]. The direction and robustness of linear relationships between latent variables and the mycorrhizal colonization of maize and the spore density of AMF were determined by the path coefficients (**S4 and S5 Tables**), and the variability or expressed variance was determined by $R^2$. Similarly, the values of weights in an SEM model allowed us to establish the links between observable variables (indicators) and latent variables. Meanwhile, the loading values were used to determine the strength of the relationship between them. Significant weights indicate that the measurements are strongly influenced by the corresponding latent variables [62].

## Results

### AMF roots colonisation of maize and spores density in the field conditions

The study sites exhibited significantly different rates of root colonization and AMF spore densities (**Fig 1**). One-way analysis of variance (ANOVA) indicated that the mycorrhizal colonization rate of maize (frequency and intensity of mycorrhization) differed significantly among the study sites (F = 25.01, *p*<0.0001 and F = 25.18, *p*<0.0001, respectively, for frequency and

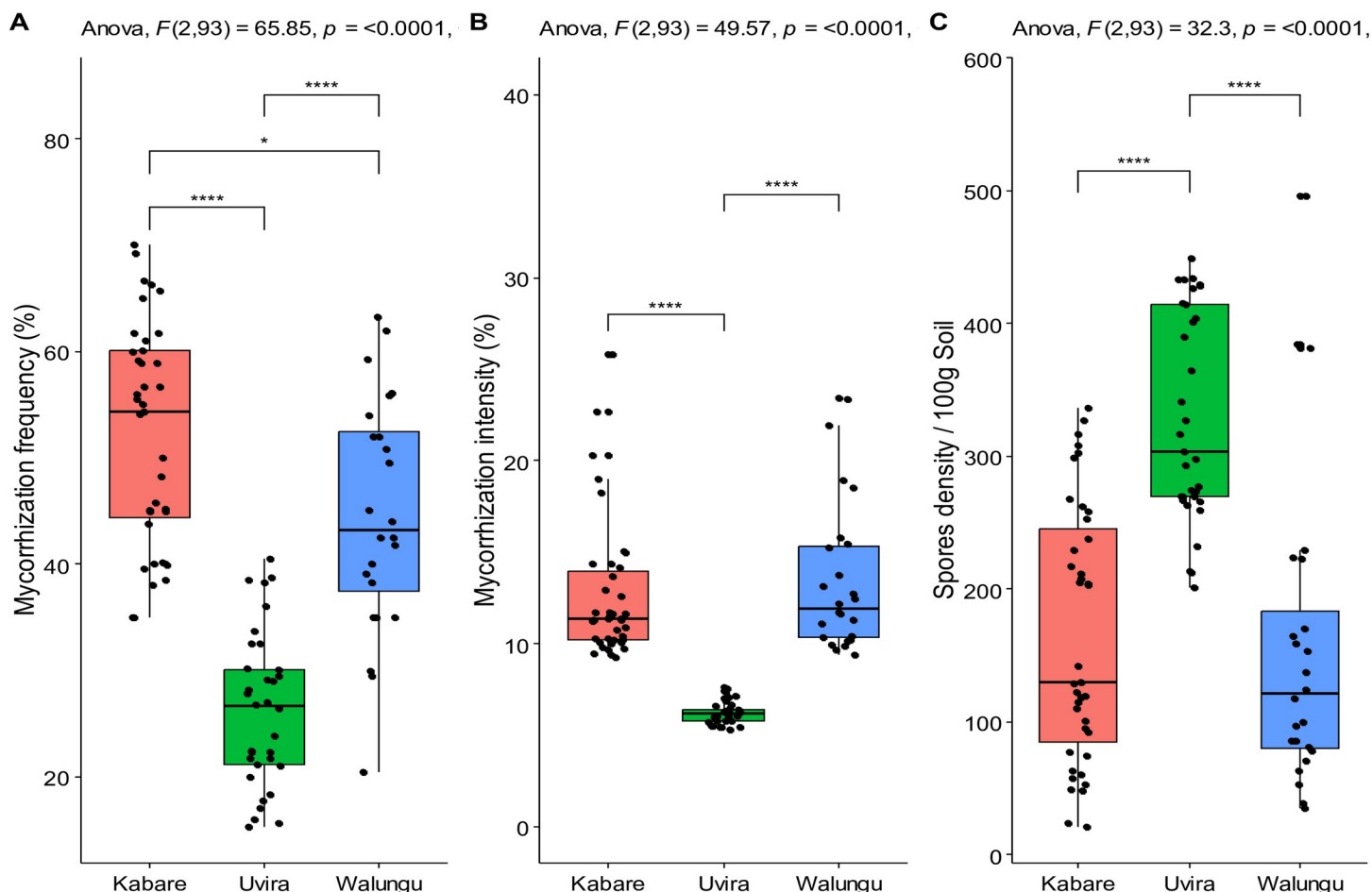

**Fig 1. Mycorrhizal Status of maize (*Zea mays* L.) in the three selected territories.** A: Frequency of arbuscular mycorrhizal, B: root colonization intensity, C: AM spore density, ANOVA: analysis of variance, ns: non-significant differences (p≥0.05), **: p<0.05 (significant), ***: p<0.01 (highly significant), and ****: p<0.0001 (very high significant.

intensity). Additionnally, there were significant differences among territories (F = 65.85, $p<0.0001$ and F = 49.57, $p<0.0001$). At the territory level, the maximum frequency of mycorrhization was observed in the territories of Kabare and Walungu (~68%), while the lowest value in Uvira (18.8%). No significant difference was observed between Walungu and Kabare, as well as between Kabare and Uvira. The intensity of maize mycorrhization significantly varied among territories, with Kabare and Walungu showing high mycorrhization intensity, while Uvira exhibited the lowest intensity of mycorrhization.

The results of the one way ANOVA indicated significant variations in AMF spore densities among territories (F = 25.16, $p<0.0001$). Remarkably, the spore density of AMF was notably higher in Uvira (329±79 spores 100 $g^{-1}$ soil) compared to other territories. No significant difference in spore density was observed between the territories of Kabare (165±26 spores100 $g^{-1}$ soil) and Walungu (156±48 spores 100 $g^{-1}$ soil). At the sampling sites level, the findings revealed substantially different mycorrhization rate across various study sites, regardless of the considered territory. Significant variations in both the frequency and intensity of natural maize mycorrhization among the study sites (F = 25.01, $p = 0.002$ and F = 25.18, $p<0.0001$, respectively) were observed.

The highest mycorrhization rates were recorded at Cirhunga Karhambi (68.3%), Bugobe cif (61.9%), Bugobe Kahave (61.1%), Bushumba (60.6%), and Bushwira center (59.2%) from Kabare in highland sites, while the lowest values were recorded at Ndunda (18.8%), Rusabagi (21.2%), Kabunambo (22.8%), Sange (22.1%), and Bwegera (22.7%), from Uvira in lowland sites (**Table 2** and **Fig 2**). Based on the results of the one-way analysis of variance, the density of AMF spores in the rhizospheric soil of maize was significantly influenced by the sites. Indeed, spore density was higher at Katogota (417 spores. 100 g-1 soil), Bwegera (431 spores. 100 $g^{-1}$ soil), and Kabunambo (381 spores. 100 $g^{-1}$ soil) from Uvira, compared to other sites (F = 128.3, $p < 0.0001$).

## Relationship between environmental parameters and the mycorrhizal status of maize under field conditions

**Environmental factors and AMF colonization of maize.** The characterization of the study area was based on climatic parameters, soil physical and chemical variables, as well as topographic parameters. According to the results of the Multiple Factor Analysis (MFA) on the various sites, the first two components (MFA1 and MFA2), taken together, explain 53.6% of the total variability in the data (**Table 3** and **Figs 3** and **S2**). The first component (MFA1), representing 31.9% of the variance, was strongly and significantly correlated (<0.05) with average, maximum, and minimum temperatures (r = 0.9, r = 0.91, r = 0.89, respectively), wind speed (r = 0.8), vapour pressure (r = 0.78), solar radiation (r = 0.91), sand content (r = 0.84), P, C, and OM content (r = 0.63, r = 0.68, r = 0.69, respectively), CTI (r = 0.57), and AMF spore

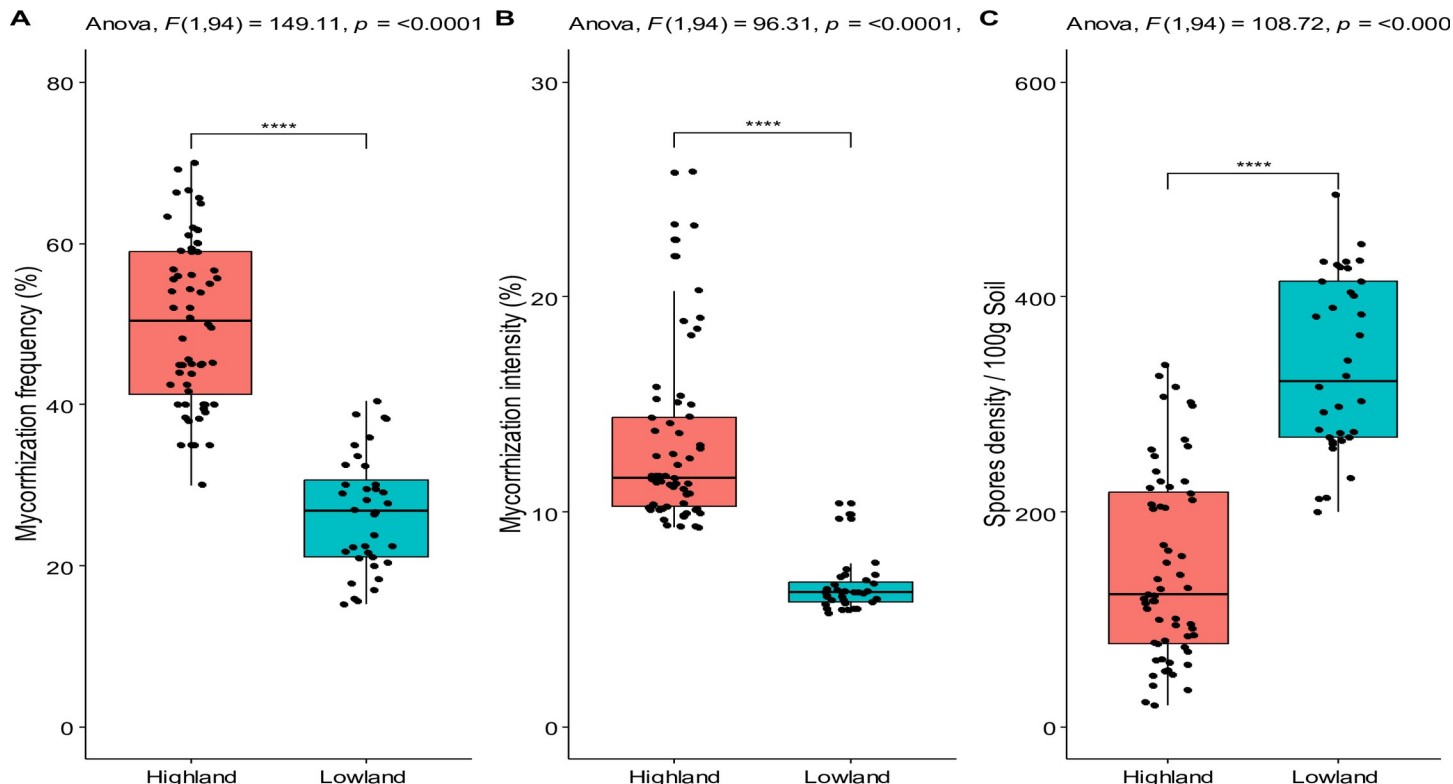

**Fig 2. Mycorrhizal Status of maize (*Zea mays* L) in the two agroecological zones (highland and lowland).** A: Mycorrhization frequency (%), B: Maize root colonisation intensity (%), C: AM spore density, ANOVA: analysis of variance, ns: non-significant differences (p≥0.05), **: p<0.05 (significant), ***: p<0.01 (highly significant), and ****: p<0.0001 (very high significant).

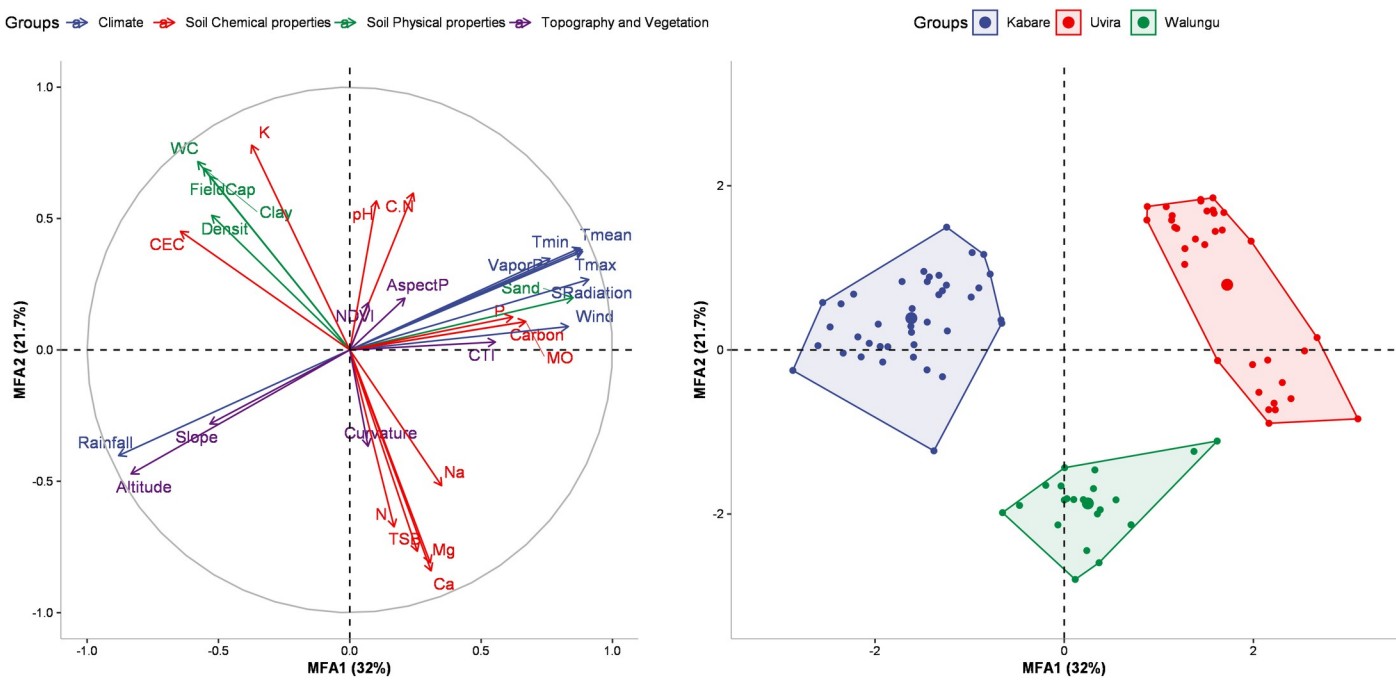

**Fig 3. Multiple factor analysis constructed based on climatic variables, physical and chemical soil properties, topographical variables, and vegetation index.** (A) Presents the projection of study variables onto the two main component MFA1-MFA2, illustrating the correlation between variables and components. (B) Displays the distribution of different study sites in the factorial plane of MFA1 and MFA2 and their grouping according to territories.

density (r = 0.56). On the other hand, precipitation (r = -0.9), CEC (r = -0.56), slope (r = -0.57), altitude (r = -0.88), and mycorrhization intensity (r = -0.41) showed an inverse effect on the first component (MFA1). The second component (MFA2) explains 21.7% of the total variance and is positively correlated with soil clay content (r = 0.7), bulk density (BD) (r = 0.58), field capacity (r = 0.72), soil water content (SWC) (r = 0.76), soil pH (r = 0.51), K content (r = 0.79), and C/N ratio (r = 0.58). On the other hand, total soil bases (TSB) (r = -0.77), N, Ca, Mg, and Na content (r = -0.65, r = -0.85, r = -0.82, r = -0.54, respectively), and mycorrhization frequency (r = -0.47) were inversely correlated with MFA2. Overall, the MFA results revealed that the three zones (territories) of Kabare, Walungu, and Uvira are significantly distinct in terms of the considered variables, with some similarities between Kabare and Walungu (**Fig 4**). According to MFA1, Uvira was observed to be different from Kabare and Walungu due to its high temperatures, solar radiation, wind speed, spore density, sand content, P, C, and OM, low precipitation, slope, CEC, maize mycorrhization rate, and low altitude. Further, it was noticed that the physical and chemical properties of the soil differentiated Kabare from Walungu along the second component (MFA2). Indeed, Kabare differs from Walungu due to high values of K, pH, C/N, field capacity, soil clay content, and soil water content, and low values of Na, Mg, TSB, N, and Ca. Similar results were observed when applying analysis of variance between sites for the various parameters under study (**Figs 4 and 5**).

The correlation analysis among the studied variables allowed us to uncover the existing relationships between the various examined parameters (**Fig 6 and S2 Table**). The results revealed that climatic characteristics and soil chemical properties are the most important variables determining maize mycorrhization. We observe that the mycorrhizal colonization (frequency and intensity) of maize is strongly, significantly, and inversely correlated with maximum, minimum, and average temperature (r = -0.78, p = 0.0003, r = -0.79, p = 0.0003, r =

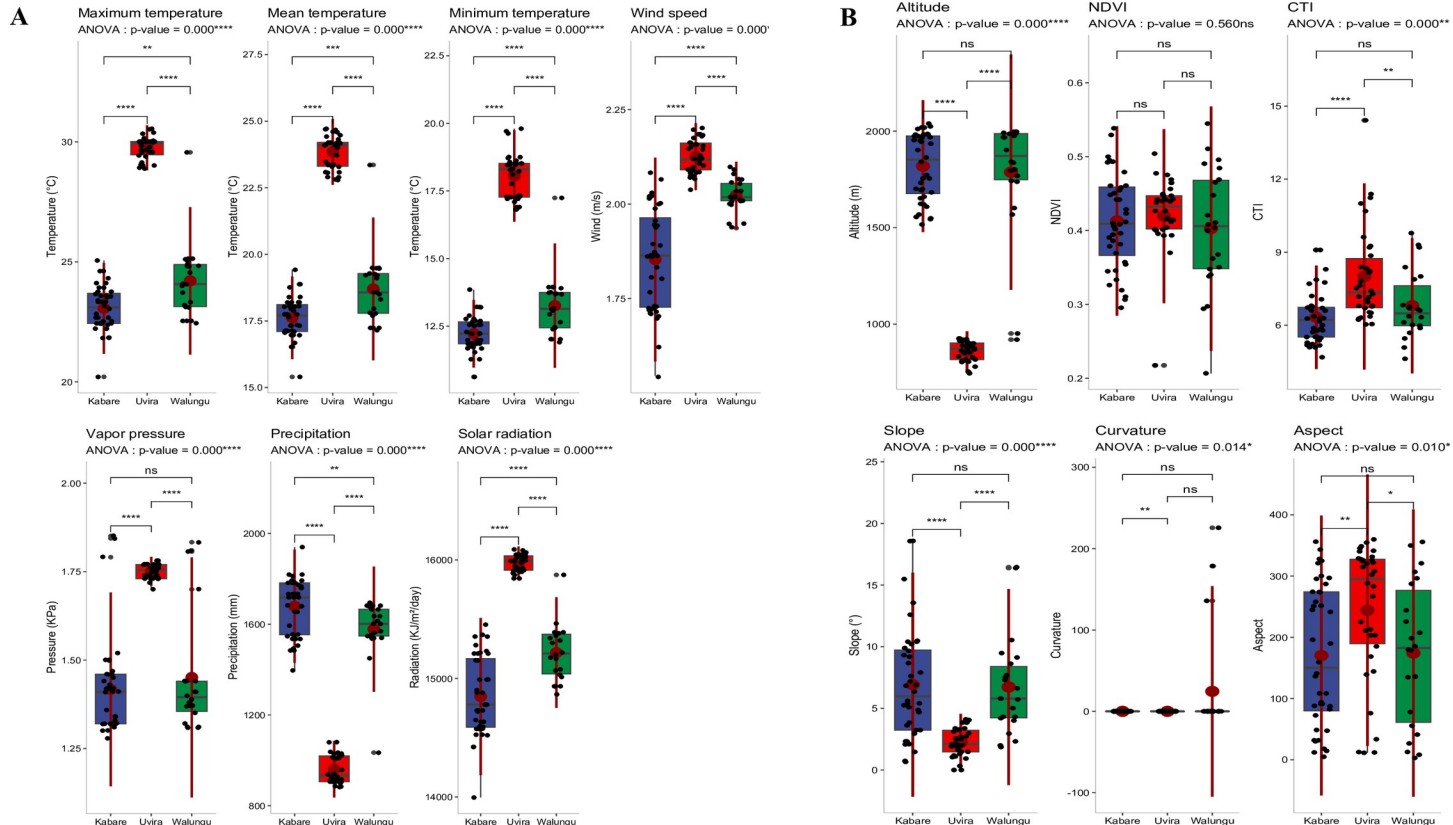

**Fig 4.** Characterization of the site climate variables (A) and topographic and vegetation indices (B) in the three territories covering the study area. The site climate parameters are average, maximum, and minimum temperatures, rainfall, wind speed, solar radiation, and vapour pressure. Topographical variables are; altitude, NDVI, CTI, slope, curvature, slope aspect. The results of the analysis of variance indicate differences between the territories; asterisks represent the level of significance, while "ns" indicates that there is no significant difference between the territories. ANOVA: Analysis of variance, ns: non-significant differences (p≥0.05), **: p<0.05 (significant), ***: p<0.01 (highly significant), and ****: p<0.0001 (very high significant).

-0.79, p = 0.0004, and r = -0.7, p = 0.003, r = -0.69, p = 0.006, r = -0.69, p = 0.003, respectively, for frequency and intensity of mycorrhization), wind speed (r = -0.64, p = 0.008, and r = -0.5, p = 0.002, respectively), solar radiation (r = -0.76, p = 0.0006, and r = -0.64, p = 0.008, respectively), P content (r = -0.51, p = 0.001, and r = -0.39, p = 0.048, respectively), C content (r = -0.67, p = 0.02, and r = -0.58, p = 0.0003, respectively), and the C/N ratio (r = -0.39, p = 0.054, and r = -0.41, p = 0.02, respectively). Furthermore, there is a positive and highly significant correlation between the mycorrhization rate and altitude (r = 0.77, p<0.0001, and r = 0.78, p<0.0001, respectively) and precipitation (r = 0.7, p = 0.01, and r = 0.79, p = 0.004, respectively). However, no significant relationship was observed between maize mycorrhization rate and TSB, ON, pH, sand content, CTI, and NDVI. According to these results, the mycorrhizal colonization of maize is determined by climatic parameters and soil chemical properties, as also indicated by the multiple factor analysis results.

Given that climatic characteristics and soil chemical properties exerted a substantial impact on maize mycorrhizal colonization, we employed the Partial Least Squares Structural Equation Modeling (PLS-SEM) approach to intricately dissect potential pathways through which climatic features, physical and chemical soil properties, as well as topographical parameters and their interplay, influence maize mycorrhizal colonization across diverse study sites. In this study, the PLS-SEM model, applied to the dataset, underwent evaluation based on variables

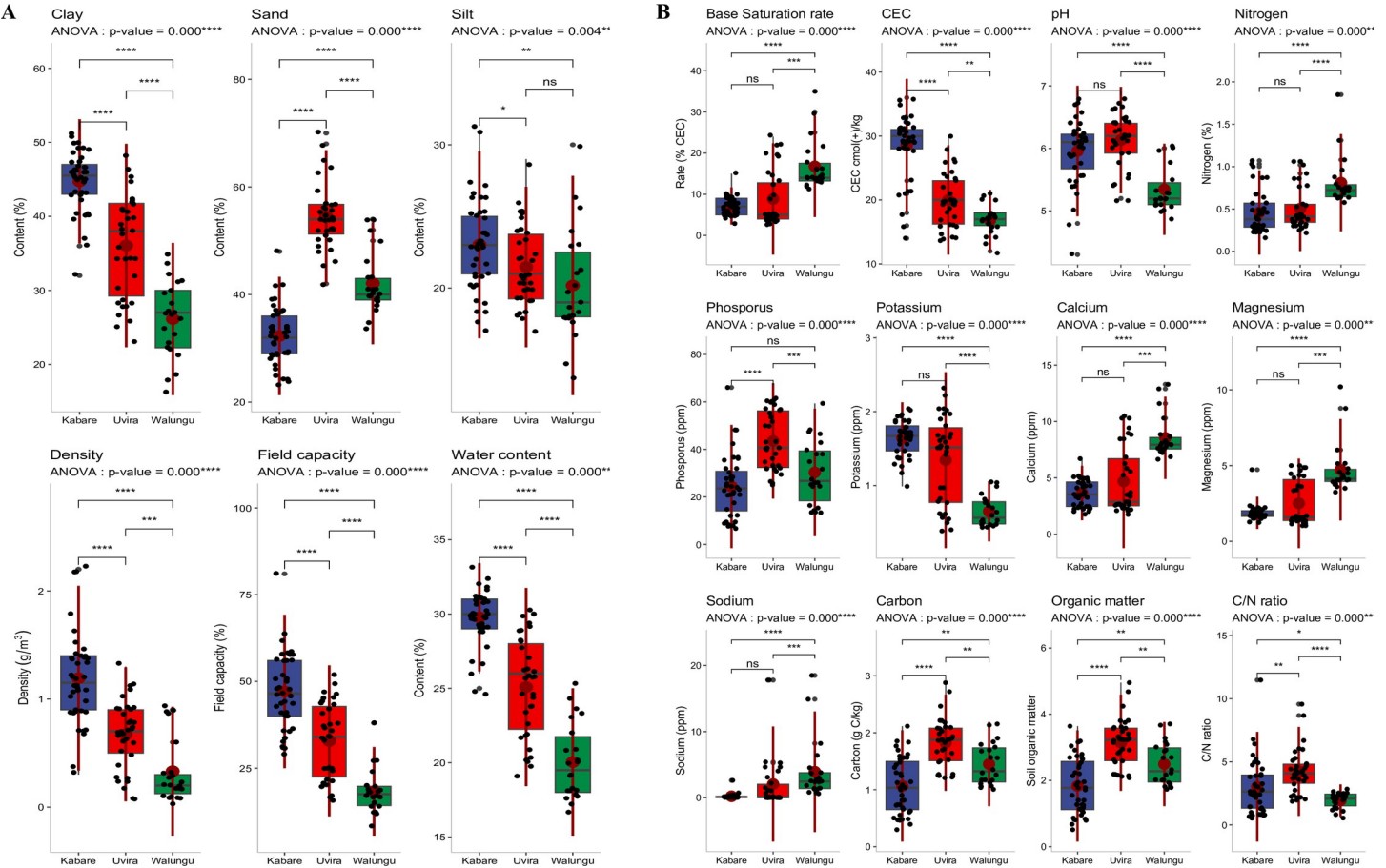

**Fig 5.** Soil physical (A) and chemical variables (B) in the three territories covering the study area. Soil physical parameters are percentage of clay content, sand content, silt content, soil density, field capacity and water content. Soil chemical properties are base saturation rate, cation exchange capacity, soil pH, N, P, K, Ca, Mg, Na, C, organic matter concentration and C/N ratio. The results of the analysis of variance indicate differences between the territories; asterisks represent the level of significance, while "ns" indicates that there is no significant difference between the territories. ANOVA: analysis of variance, ns: non-significant differences ($p \geq 0.05$), **: $p < 0.05$ (significant), ***: $p < 0.01$ (highly significant), and ****: $p < 0.0001$ (very high significant).

such as weights, path coefficients, and the coefficient of determination, as depicted in the **S3 Table**. It was discerned that the chosen PLS-SEM model elucidated a robust variance ($R^2$ = 0.77) in maize mycorrhizal colonization.

The PLS-SEM model delineates that site climate (primarily Altitude) and soil chemical properties (C, P, and TSB) stand out as the principal latent variables significantly shaping maize mycorrhizal colonization (**Figs 7 and S3**). Nevertheless, the direct impact of site climate (path coefficient = -0.63) surpassed that of soil chemical properties (path coefficient = -0.31). Notably, the outcomes suggest that altitude and TSB wielded a direct, positive, and statistically significant influence on the natural colonization of maize by AMF (weight of -0.99 for Altitude and -0.56 for TSB). Conversely, variables C (weight) and P (weight) exhibited a noteworthy and inverse effect on maize root colonization by AMF.

**Environmental factors and spore density in maize rhizosphere.** The PLS-SEM model developed to investigate the causal effects between latent variables and the density of AMF spores revealed a high variance ($R^2$ = 0.69) in the total AMF spores density across the study sites (**Figs 8 and S4 and S4 Table**). According to this model, the site climate (indicators: Tmean, Tmax, and Tmean) and soil physical properties (indicators: Clay, Field capacity, and

**Table 3. Correlations between soil physico-chemical properties, site climatic parameters, AMF spores density and mycorrhizal root colonization of maize and MFA axes (The bold values indicate significant correlations).**

| Variables | Dim1 | Dim2 | Dim3 |
|---|---|---|---|
| Variance | 2.44 | 1.65 | 0.77 |
| Percentage of variance | 31.9 | 21.7 | 10.2 |
| Cumulative of % of variance | 31.9 | 53.6 | 63.8 |
| **Site climate** | | | |
| Tmax | **0.91** | 0.240 | 0.23 |
| Tmean | **0.90** | 0.247 | 0.23 |
| Tmin | **0.89** | 0.254 | 0.24 |
| Wind | **0.80** | -0.036 | 0.26 |
| VaporP | **0.78** | 0.251 | 0.19 |
| Rainfall | **-0.90** | -0.262 | -0.22 |
| SRadiation | **0.91** | 0.128 | 0.23 |
| **Soil physical properties** | | | |
| Clay | -0.49 | **0.701** | 0.12 |
| Sand | **0.84** | 0.07 | 0.29 |
| Soil density | -0.44 | **0.58** | -0.14 |
| FieldCap | -0.45 | **0.72** | 0.02 |
| WC | -0.49 | **0.76** | 0.03 |
| **Soil Chemical properties** | | | |
| TSB | 0.19 | **-0.77** | -0.08 |
| CEC | **-0.56** | 0.53 | -0.21 |
| pH | 0.14 | **0.51** | 0.24 |
| N | 0.11 | **-0.65** | -0.07 |
| P | **0.63** | 0.03 | 0.09 |
| K | -0.31 | **0.79** | 0.09 |
| Ca | 0.24 | **-0.85** | -0.12 |
| Mg | 0.21 | **-0.82** | 0.03 |
| Na | 0.29 | **-0.54** | -0.13 |
| Carbon | **0.68** | 0.07 | 0.12 |
| MO | **0.69** | 0.07 | 0.12 |
| C.N | 0.29 | **0.58** | 0.21 |
| **Topographic parameters** | | | |
| NDVI | 0.15 | 0.22 | **-0.65** |
| CTI | **0.57** | -0.04 | -0.39 |
| Slope | **-0.54** | -0.17 | 0.14 |
| Curvature | 0.02 | -0.31 | -0.03 |
| AspectP | 0.18 | 0.17 | **0.69** |
| Altitude | **-0.88** | -0.35 | -0.11 |
| **Mycorrhization parameters** | | | |
| Frequency | -0.25 | **-0.47** | 0.31 |
| Intensity | **-0.41** | **-0.41** | 0.36 |
| Spores density | **0.561** | -0.10 | -0.35 |

Sand) are the main latent variables explaining the variability in AMF spore density (path coefficients = 0.59 and -0.48, respectively) in the maize rhizosphere in different study sites. Indeed, we observed that temperatures Tmean (loading of 0.72), Tmin (loading of 0.71), and Tmax (loading of 0.68) had a significant and positive impact on AMF spore density in the different

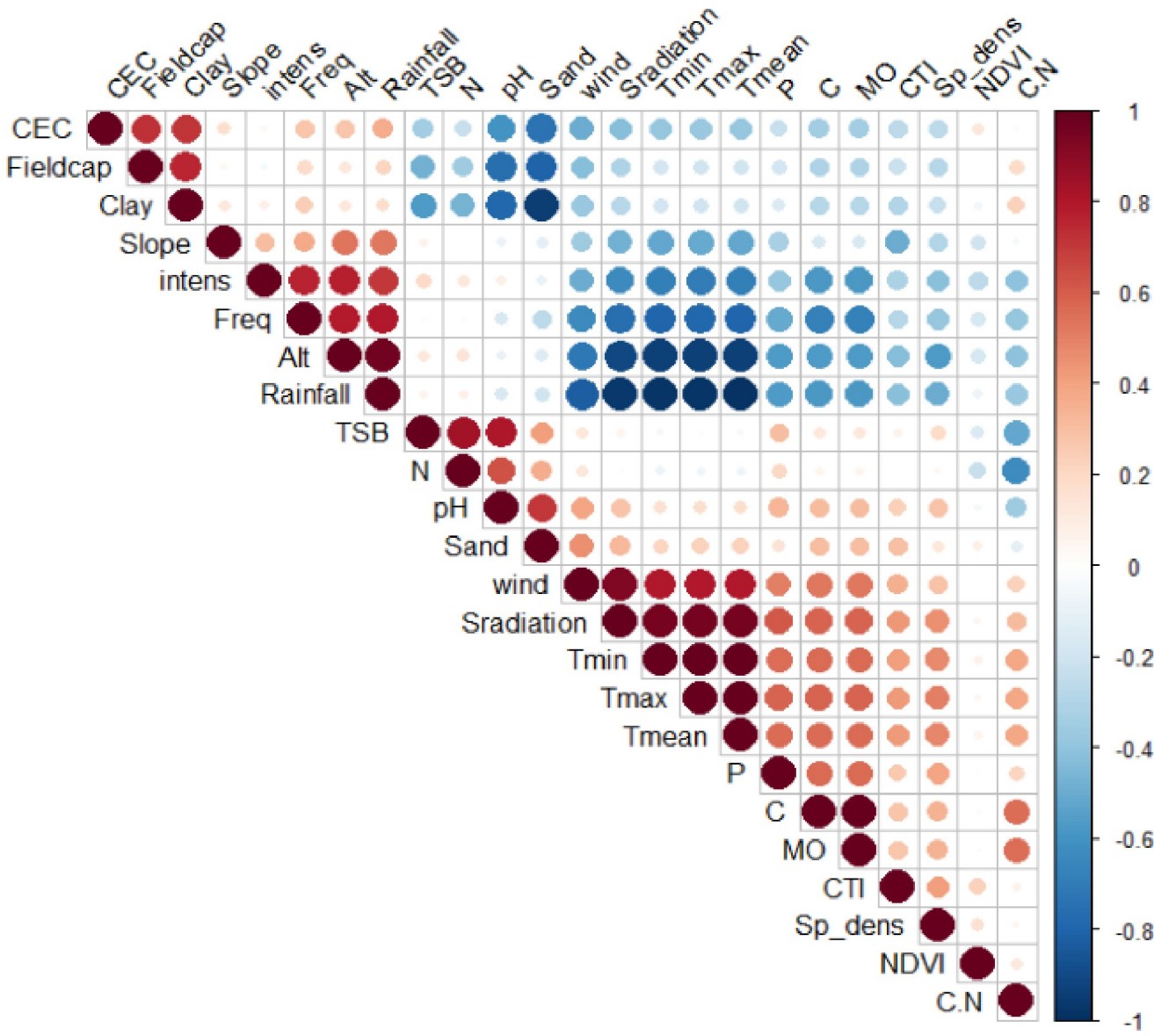

**Fig 6. Pearson correlation matrix among different groups of variables, including site climate characteristics (average, maximum, and minimum temperatures, rainfall, wind speed, solar radiation, and vapour pressure), soil physical properties (clay content, sand content, silt content, soil density, field capacity, and water content), soil chemical properties (cation exchange capacity, soil pH, N, P, K, Ca, Mg, Na, C, organic matter concentration, and C/N ratio), and topography and vegetation (altitude, NDVI, CTI, slope, curvature, slope aspect).**

sites. On the other hand, soil physical properties such as clay, field capacity, and sand were significantly and inversely associated with AMF spore density in the maize rhizosphere (weights of 2.25, 1.06, and 2.71, respectively, for clay, field capacity, and sand).

Furthermore, the Pearson correlation analysis between AMF spore density and environmental parameters revealed that AMF spore density is also explained by climatic characteristics and soil chemical properties (S5 Table). Indeed, the density of AMF spores is significantly and inversely correlated with rainfall (r = -0.5, p = 0.003) and altitude (r = -0.56, p = 0.0005). On the other hand, a positive and significant relationship was observed between AMF spore

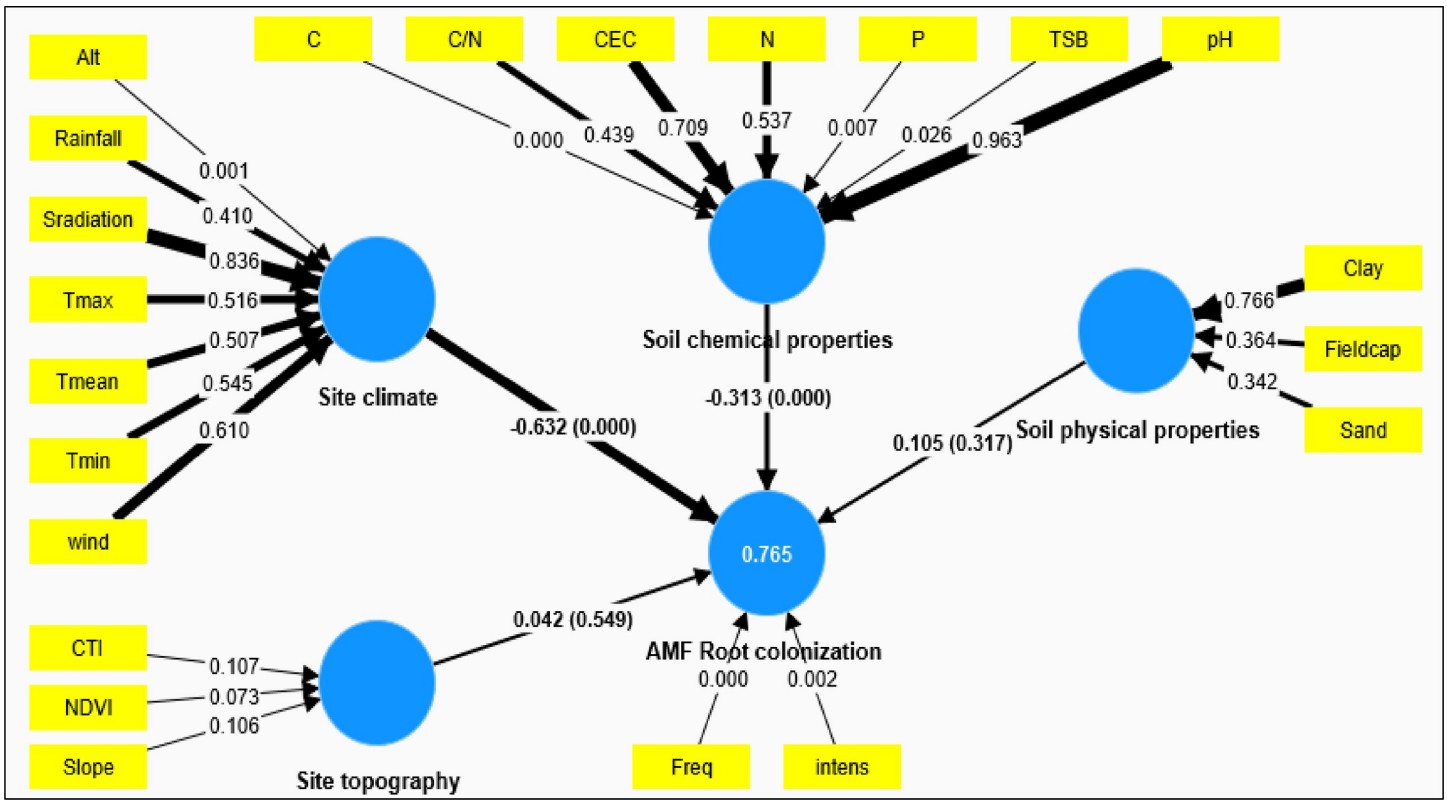

**Fig 7. Partial least square structural equation model (PLS-SEM) showing the direct effect of site climate characteristic, soil physicochemical properties and the site topography on the mycorhization of maize in the field condition.** (R2 = 0.77, CR = 1 and AVE = 1). Bold lines and non-bold lines indicate significant and non-significant pathways of the latent variables, respectively. The figures alongside the arrows for latent variables denote the path coefficients, with the values in parentheses indicating the corresponding p-values. For the indicators of latent variables, the numbers presented alongside the arrows represent the p-values. freq: mycorrhizal frequency, Intens: Mycorrhizal intensity, CEC: cation exchange capacity, Fieldcap: Field capacity, Clay: soil clay content, Alt: altitude, TSB; base saturation rate, N: soil nitrogen content, pH: soil pH, Sand: soil sand content, Wind: wind speed, Srad: solar radiation, Tmin: minimum temperature, Tmax:maximum temperature, Tmean: mean temperature, P:, C:, MO: organic matter, AM spore density, CTI:, NDVI:, C.N: C. ratio.

density and solar radiation (r = 0.45, p = 0.01), and CTI (r = 0.41, p = 0.02), suggesting an increase in spore density as these parameters increase.

## Discussion

### Natural mycorrhizal colonization in maize

The results of this study revealed a significant and natural colonization of maize roots by indigenous AMF in the various surveyed sites. The observed rate of mycorrhizal colonization of maize showed significant variations among sites, ranging from 18.8% to 68.7%. Higher rates of mycorrhization were observed in the territories of Kabare and Walungu compared to Uvira. This study provides initial insights into the natural mycorrhization of maize in South Kivu. Other researchers have reported that maize naturally develops mycorrhizal symbiosis with different indigenous soil strains without inoculation [63, 64]. The elevated rates of mycorrhizal colonization in maize could be linked to the plant's high demand for nutrients in the region, thus promoting optimal growth and grain development. Indeed, when a plant has an increased nutrient demand, it may foster a more substantial mycorrhizal colonization to enhance the absorption of essential soil nutrients [34]. Conversely, the low mycorrhization rate of maize in different sites could be associated with soil physicochemical properties, such as soil moisture

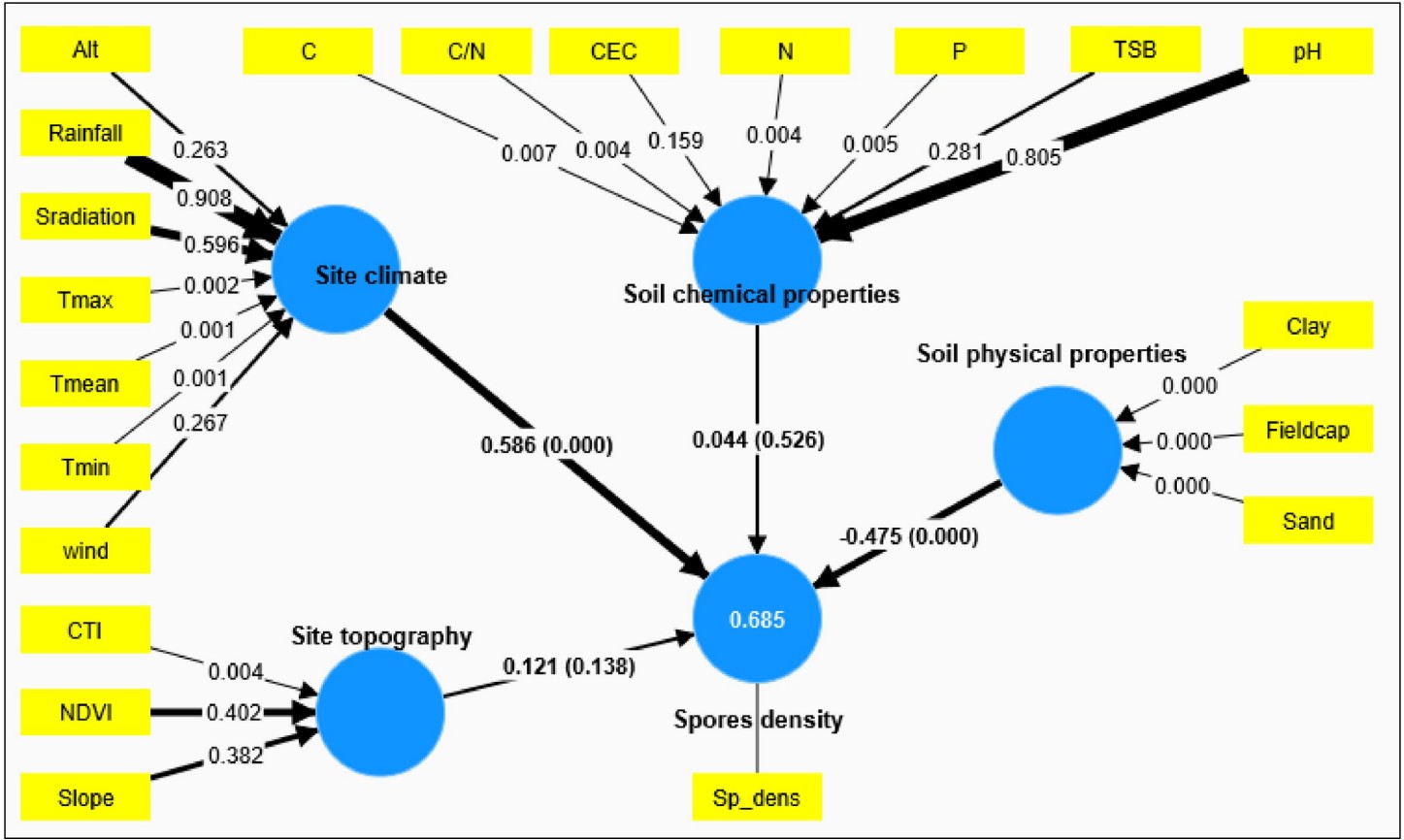

**Fig 8. Partial least square structural equation model (PLS-SEM) showing the direct effect of site climate characteristic, soil physicochemical properties and the site topography on the AMF spore density in the field condition.** $R^2 = 0.685$, CR = 1 an AVE = 1. Bold lines and non-bold lines indicate significant and non-significant pathways of the latent variables, respectively. The figures alongside the arrows for latent variables denote the path coefficients, with the values in parentheses indicating the corresponding p-values. For the indicators of latent variables, the numbers presented alongside the arrows represent the p-values.: Sp_dens: spores density, CEC: cation exchange capacity, Fieldcap: Field capacity, Clay: soil clay content, Alt: altitude, TSB; base saturation rate, N: soil nitrogen content, pH: soil pH, Sand: soil sand content, Wind: wind speed, Srad: solar radiation, Tmin: minimum temperature, Tmax:maximum temperature, Tmean: mean temperature, P:, C:, MO: organic matter, AM spore density, CTI:, NDVI:, C.N: C/N ratio.

and varying nutrient concentrations. This result confirms our initial hypothesis that maize mycorrhization rates would vary depending on the sites and agroecological zones. It has been demonstrated that maize mycorrhization can significantly vary based on the availability of elements in the soil, such as carbon and nitrogen, and may be constrained by high phosphorus concentrations [34, 63, 65].

The low rate of maize mycorrhization in different sites could also be attributed to the low compatibility between the cultivated maize genotype and the indigenous strains of Arbuscular Mycorrhizal Fungi (AMF) colonizing it. Indeed, different host plant genotypes may preferentially develop mycorrhizal symbiosis with specific AMF species [66, 67]. However, in this study, we did not consider the varieties used by farmers in the study sites. Similarly, the effectiveness of mycorrhizal colonization varies from one AMF species to another [68].

### Site climate and soil properties effects on mycorrhizal colonization

Among all the climatic parameters considered in this study, the results highlighted a positive and significant correlation between rainfall and mycorrhizal colonization of maize in different sites. Indeed, higher rainfall improves soil moisture, creating favourable conditions for spore

germination in the soil and the colonization of the roots of the host plant [69]. Additionally, drought conditions negatively impact maize development, reducing its photosynthetic capacity and mycorrhizal colonization rate [70]. This confirm our second hypothesis that climate variable could explain the variability of the natural mycorhization of maize in the field. Previous studies have demonstrated that precipitation plays a crucial role in determining mycorrhization due to its influence on the composition of AMF communities and mycorrhizal colonization in various species [71]. Indeed [72], reported a positive correlation between precipitation and the mycorrhizal colonization of *Ilex paraguariensis* roots in their study. According to these authors, there was an observed increase in mycorrhization rates ranging from 12% to 19% during the high precipitation season compared to the low precipitation season. Similarly [73], demonstrated a direct association between higher precipitation levels and increased vesicle production. Our findings also revealed that other climatic parameters, such as average, minimum, and maximum temperature, wind speed, and solar radiation, exhibit an inverse correlation with the mycorrhizal colonization of maize. This aligns with prior research [3, 8], which has indicated that a decrease in mycorrhizal colonization is induced by thermal stress. This thermal effect could be linked to changes in the internal and external structures of the fungus, consequently impacting the structure of the arbuscular mycorrhizal fungi (AMF) hyphal network and resulting in a reduction in vesicles and fungal hyphae [3]. Furthermore, under high-temperature conditions, a decrease in the photosynthetic capacity of the host plant is observed, leading to a reduction in carbon allocation to arbuscular mycorrhizal fungi (AMF) and, consequently, to low mycorrhization [74]. Moreover, reports indicate that the effects of temperature on mycorrhization vary depending on the host plant and the temperature range considered [29]. Indeed, under certain conditions, temperatures exceeding 23°C are more conducive to sporulation than to root colonization by AMF [75]. However, other research has shown that the increase in temperature between 25 and 40°C does not affect mycorrhizal colonization [76], while under these same conditions [31], observed a significant reduction in mycorrhizal colonization in sorghum.

It also appears that the impact of climatic factors on maize mycorrhizal colonization and spore density varies along the altitudinal gradient. Our results indicate that maize root colonization is positively correlated with altitude, while spore density is negatively correlated with altitude. Although altitude is not directly a climatic parameter, it nonetheless has a direct and significant influence on various climatic parameters such as temperature, precipitation, solar radiation, etc., and consequently affects maize mycorrhization [77]. However, other studies suggest that mycorrhizal colonization may be higher at lower altitudes compared to higher altitudes [78]. This discrepancy with these studies could be attributed to other factors such as soil properties and environmental topography. In this study, we applied the PLS-SEM model to better understand the complexity of the relationship between climatic parameters, physical and chemical soil properties, and site topography concerning maize mycorrhizal colonization and spore density.

The PLS-SEM model revealed that maize mycorrhizal colonization is influenced by site climate and soil chemical properties. The climate effect was supported by a positive correlation between mycorrhizal colonization and altitude, ranging from 745 to 2040 meters above sea level across the different study sites. These findings align with those reported by [79], who investigated the impact of altitude on mycorrhizal symbiosis and functional diversity of mycorrhizal symbiosis in gradients of altitude almost similar to ours (<4000 meters above sea level). The direct effect of altitude on natural maize mycorrhization is likely due to its direct influence on the climatic parameters of the sites, which, in turn, determine the establishment of mycorrhizal symbiosis in maize [77]. [80] reported high rates of mycorrhizal colonization in *Pennisetum centrasiaticum* species in high-altitude mountain forests. Similarly [81], also

reported abundant mycorrhizal colonization in the roots of perennial grasses at high altitudes. Our SEM model showed that altitude negatively affected the density of AMF spores in the maize rhizosphere. One possible explanation for this situation could be that altitude creates favourable conditions for the growth of the host plant, spore germination of AMF, and therefore is conducive to root colonization rather than spore formation at medium and high altitudes compared to low altitude [82]. According to Giovannetti et al. [83] the favourable conditions for spore germination of most AMF species include relatively low temperatures (between 10 and 25°C), good soil moisture around field capacity, and an adequate soil nutrient content. However, germination is inhibited by elements such as inorganic phosphorus, heavy metals, soil acidity, and salinity. On the other hand [27, 84], reported an opposite effect of altitude on the mycorrhization of plant species in wetlands and in wheat on dry land, as well as a positive effect on spore abundance. This trend of altitude on mycorrhization could be related to the fact that, in their studies, the authors considered extreme altitude values (between 4000 and 5000 m and between 2 and 649 m), for which maize cultivation is practically not feasible, compared to the altitude range considered in this study. Another explanation could be that mycorrhizal colonization at different altitudes depends on the host plant and its responsiveness to mycorrhization under these conditions [85, 86].

We highlighted partial support for our hypothesis that soil chemical properties (higher nutrients level, especially available P and C) have a negative effect on natural maize root colonization by AMF in the field soil. According to the PLS-SEM model, the impact of soil chemical properties on maize mycorrhizal colonization was confirmed by a positive correlation with TSB and a negative correlation with C and P content. The influence of basic soil cations on mycorrhizal colonization could be linked to their buffering effect, reducing soil acidity and enhancing the photosynthetic capacity of the host plant [87]. [88] demonstrated a significant increase in mycorrhizal colonization and photosynthesis in *Acer saccharum* in response to basic cation amendments. In contrast, several studies have reported a negative influence of P on maize mycorrhizal colonization [89, 90]. It has indeed been shown that a high concentration of soil nutrients, such as phosphorus, significantly reduces fungal species diversity in the soil, root colonization, and the efficiency of mycorrhizal symbiosis in maize [91, 92]. Inded, it is established that, at the flowering stage of *Vigna unguiculata* (L.), the frequency and intensity of mycorrhizal colonization is negatively correlated with available P, which explain about 78% of the variation in those variables [93]. As for carbon, the negative influence on maize mycorrhizal colonization could be attributed to the fact that carbon (organic matter) serves as an available energy source for fungi, limiting the initiation of mycorrhizal symbiosis, given that AMF is well-endowed with energy [94]. These results imply that plants become less dependent on mycorrhization in highly fertile soil, which also leads to lower AMF activity [95]. However [96], reported a positive correlation, while Ndeko et al. [19] did not find a significant relationship between common bean mycorrhizal colonization and soil carbon concentration. In their study, Liu et al. [95] showed that high soil carbon concentrations significantly reduce mycorrhizal colonization of maize, as well as the glomaline related soil protein content.

## Site climate and soil properties effects on AMF spores' density

Our study also aimed to identify the determinants of AMF spore density in the rhizospheric soil of maize among the various environmental parameters investigated. The high abundance of AMF spores in certain study areas could be attributed to a strong sporulation capacity linked to anthropogenic factors as well as local environmental conditions [48, 84]. Our results are consistent with those reported by [38], who demonstrated some variability in the number of AMF spores across sites in the same study area. However, their findings differ from ours in

that they reported low spore densities in maize fields. This difference could be related to the limited number of sites considered in their study and the timing and sampling method employed. We have established an inverse correlation between spore density and altitude. This relationship can be explained by the influence of altitude on climate, leading to notable variations in climatic variables, land use, and vegetation from low to high altitudes (lowland and highland), particularly in eastern DRC. Thus, the transition from low to high altitudes could create conditions more conducive to the sporulation of AMF, explaining the high sporulation observed at lower altitudes. For example, the study by [81, 97] reported that AMF spore number in the rhizosphere of *Siraitia grosvenorii* is negatively correlated with altitude. Similarly [48, 98], reported that spore density decreases with elevated temperature, which tends to increase at lower altitudes in South Kivu. The results of the SEM model have shown that the temperature and clay concentration, sand concentration, and field capacity, considered together, significantly affect the AMF spores density. AMF spores density has was inversely correlated with rainfall and a positively with minimum, maximum, and average temperatures. Indeed, the AMF spores density is associated with three processes in the soil, namely, spore germination, hyphal sporulation, and spore mortality. Thus, it appears that these processes are significantly influenced by precipitation and temperature [82]. In the literature, several studies support the idea that rainfall is linked to high soil water content, which would reduce sporulation and negatively impact spore density due to the aerobic nature of AMF [87]. Others argue that optimal soil moisture, not excess, would favour the development of the host plant, mycorrhizal colonization, and limit the sporulation process [99]. The positive effect of temperature on spore density is primarily related to its influence on the formation of AMF spores and, consequently, the spore number. Given that the ideal temperature for mycelium development varies between 20 and 30°C, an increase in temperature beyond this range will promote spore formation [75].

The abundance of spores and the distribution of mycorrhizal communities depend on various factors related to the local environment [82]. In this study, spore density is inversely correlated with the clay content, sand content, and field capacity of the soil (PLS-SEM model). These results are consistent with those reported by [48], who investigated the relationships between soil characteristics and spore density in the rhizospheric soil of rice. According to these authors, the clay content of the soil significantly reduces rice mycorrhization and spore density. The importance of soil clay content in determining spore density is attributed to the fact that a high soil clay concentration promotes soil compaction, which negatively impacts spore viability [100]. The negative influence of sandy texture on spore density has been reported in previous studies [94]. However, other researchers have reported that sandy soil texture tends to promote spore formation [101]. This study demonstrated that the climate characteristics of the site and the physicochemical properties of the soil explain up to 76.5% and 68.5% of the variance in natural maize mycorrhization and associated spore density in different study sites. Very recently, it has been established that a high abundance of AMF spores in agricultural soils is linked to an increased sporulation of indigenous AMF species in response to frequent disturbances in these environments [94].

However, the variability of maize mycorrhization and spore density could significantly differ depending on the genotypes used and the inoculated AMF strains [22, 102, 103]. In this study, we did not consider the response of maize genotypes to mycorrhization under different environmental conditions. Therefore, future research could focus on assessing the effect of the interaction between maize genotypes and environmental factors on the establishment of mycorrhizal symbiosis and the responsiveness of maize to mycorrhization. Furthermore, subsequent studies should identify, characterize, and test indigenous AMF strains in different sites and explore the variability in response to mycorrhizal inoculation in maize, considering both

local and improved varieties. Additionally, it will be essential to develop statistical models to predict maize mycorrhizal colonization and spore density based on climatic factors and soil physicochemical properties.

## Conclusion

This research reveals a significant variation in the natural mycorrhization of maize influenced by environmental factors in the rainy conditions of South Kivu. The results indicate that soil chemical properties and the site's climatic characteristics significantly influence the mycorrhizal colonization of maize roots, while soil physical properties do not have a substantial impact. The PLS-SEM model demonstrates that the natural colonization of maize roots by AMF is more strongly explained by the site's climatic factors than by soil chemical properties. Additionally, the site's climate, rather than soil physical and chemical properties, significantly impacts the spore density of AMF in maize cropland in South Kivu. This study represents the first comprehensive investigation into the natural mycorrhizal status of maize and the impact of environmental factors on natural mycorrhization and spore density under the rainy conditions of South Kivu. Further research is required to characterize the diversity and select indigenous AMF strains adapted to these conditions, which play a crucial role in enhancing maize productivity and mitigating the impact of conventional farming practices in rainfed agricultural systems in eastern DRC..

## Supporting information

**S1 Table. Descriptive statistics of climatic parameters, physical and chemical soil properties, and topographical and vegetation parameters used, alongside mycorrhization parameters.**
(DOCX)

**S2 Table. Pearson correlation matrix (correlation coefficient R) illustrating the relationships between climatic features, physical and chemical soil properties, topographic parameters, mycorrhizal colonization of maize, as well as the density of AMF.**
(DOCX)

**S3 Table. PLS-SEM model results for the effect of predictor variable on maize roots colonization showing the weight, loading values, and the path coefficients of the model and their associated p-values.**
(DOCX)

**S4 Table. PLS-SEM model results for the effect of predictor variable on AMF spore density in maize rhizosphere showing the weight, loading values, and the path coefficients of the model and their associated p-values.**
(DOCX)

**S5 Table. Identification of sites, climatic and soil data collected at sampling points in the two agroecological zones of South Kivu.**
(PDF)

**S1 Fig. Methodological framework for data collection, treatment and evaluating the impact of environmental factors on natural mycorrhization in maize and spore density under field conditions.**
(TIF)

**S2 Fig. Heat map analysis of interactions between climatic factors, soil physicochemical properties, and topographic and vegetation characteristics.**
(TIF)

**S3 Fig. Histograms illustrating the total effects of various latent variables on the natural mycorrhizal colonization of maize by endogenous strains of CMA derived from the PLS-SEM model.**
(TIF)

**S4 Fig. Histograms illustrating the total effects of various latent variables on the AMF spores density in maize rhizosphere derived from the PLS-SEM model.**
(TIF)

## Acknowledgments

We are grateful to the farmers for their cooperation and valuable assistance during the collection of samples from their respective fields. The authors would like to thank the "Laboratoire Commun de Microbiologie, LCM, IRD-ISRA-UCAD, Senegal" for their support during the experimental set-up phase.

## Author Contributions

**Conceptualization:** Adrien Byamungu Ndeko, Abdala Gamby Diedhiou, Saliou Fall, Gustave Nachigera Mushagalusa, Aboubacry Kane.

**Data curation:** Adrien Byamungu Ndeko, Géant Basimine Chuma, Yannick Mugumaarhahama, Aboubacry Kane.

**Formal analysis:** Adrien Byamungu Ndeko.

**Funding acquisition:** Adrien Byamungu Ndeko, Saliou Fall, Gustave Nachigera Mushagalusa, Aboubacry Kane.

**Investigation:** Adrien Byamungu Ndeko.

**Methodology:** Adrien Byamungu Ndeko, Gustave Nachigera Mushagalusa, Aboubacry Kane.

**Project administration:** Adrien Byamungu Ndeko, Gustave Nachigera Mushagalusa, Aboubacry Kane.

**Resources:** Adrien Byamungu Ndeko, Gustave Nachigera Mushagalusa, Aboubacry Kane.

**Software:** Adrien Byamungu Ndeko.

**Supervision:** Abdala Gamby Diedhiou, Hassna Founoune-Mboup, Diegane Diouf, Saliou Fall, Gustave Nachigera Mushagalusa, Aboubacry Kane.

**Validation:** Adrien Byamungu Ndeko, Abdala Gamby Diedhiou, Saliou Fall, Gustave Nachigera Mushagalusa, Aboubacry Kane.

**Visualization:** Adrien Byamungu Ndeko.

**Writing – original draft:** Adrien Byamungu Ndeko.

**Writing – review & editing:** Adrien Byamungu Ndeko, Abdala Gamby Diedhiou, Hassna Founoune-Mboup, Géant Basimine Chuma, Diegane Diouf, Saliou Fall, Gustave Nachigera Mushagalusa, Aboubacry Kane.

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
