## [Decision Letter · Decision Letter 0]

22 Jul 2024

PONE-D-24-15543Site climate more than soil properties and topography shape the natural arbuscular mycorrhizal symbiosis in maize and spore density within rainfed maize (Zea mays L.) cropland in the Eastern DR CongoPLOS ONE

Dear Dr. Byamungu,

Thank you for submitting your manuscript to PLOS ONE. After careful consideration, we feel that it has merit but does not fully meet PLOS ONE’s publication criteria as it currently stands. Therefore, we invite you to submit a revised version of the manuscript that addresses the points raised during the review process.

We look forward to receiving your revised manuscript.

Kind regards,

Marcela Pagano, Ph.D, M.D.

Academic Editor

PLOS ONE

Journal Requirements:

4. We note that [Figure 1 and Supporting Information Fig 1] in your submission contain [map/satellite] images which may be copyrighted. All PLOS content is published under the Creative Commons Attribution License (CC BY 4.0), which means that the manuscript, images, and Supporting Information files will be freely available online, and any third party is permitted to access, download, copy, distribute, and use these materials in any way, even commercially, with proper attribution. For these reasons, we cannot publish previously copyrighted maps or satellite images created using proprietary data, such as Google software (Google Maps, Street View, and Earth). For more information, see our copyright guidelines: http://journals.plos.org/plosone/s/licenses-and-copyright.

a. You may seek permission from the original copyright holder of Figure 1 and Supporting Information Fig 1 to publish the content specifically under the CC BY 4.0 license.  

Additional Editor Comments:

The manuscript was improved; however minor details are needed:

--Figure 3: Mycorrhizal Status of maize (Zea mays L) in the three selected territory

--Replace by in the three selected territories or better, use <sites>

Reviewers' comments:

Reviewer's Responses to Questions

**Comments to the Author**

1. Is the manuscript technically sound, and do the data support the conclusions?

Reviewer #1: Yes

Reviewer #2: Yes

2. Has the statistical analysis been performed appropriately and rigorously? 

Reviewer #1: Yes

Reviewer #2: Yes

3. Have the authors made all data underlying the findings in their manuscript fully available?

Reviewer #1: Yes

Reviewer #2: Yes

4. Is the manuscript presented in an intelligible fashion and written in standard English?

Reviewer #1: Yes

Reviewer #2: Yes

5. Review Comments to the Author

Reviewer #1: This study investigated the impacts of site climate, soil properties, and topography on the natural arbuscular mycorrhizal symbiosis in maize and spore density within rainfed maize (Zea mays L.) cropland in the Eastern DR Congo. The data were harvested from the 32 sites across three territories. The study employs a well-thought-out experimental design, encompassing site climate variables, soil physicochemical properties, topography, and vegetation variables. This design aids in analyzing the influence of different environmental factors on maize MC and the density of AMF spores and provides reliable data to support the study's conclusions. The article provides a detailed interpretation of the experimental results, highlighting the positive role of site climate factors on arbuscular mycorrhizal fungi colonization and spore density of maize rhizosphere soil. The explanation of the results is clear and well-structured, making it easy for readers to comprehend the study's findings. The authors mention the potential value of the research results in practical field applications. This connection to real-world applications is helpful in translating research findings into actionable recommendations for agricultural production in rainfed maize cropland systems. It is clear that the authors are not concerned about the latest publications in this field in the sections of Introduction and Discussion. The presentations are not sufficient to clarify the key issue and background in terms of AMF colonization. The latest publications might be considered as Rehman et al., 2022, Plant Soil, 1–17; Khan et al., 2024, Science of The Total Environment, 917, 170417, and Abrar et al., 2024, Plant Physiology and Biochemistry, 213, 108839.

In summary, this manuscript has several positive attributes, but minor improvements could further enhance the research.

1) Lines 27: Units of spores density should be uniform.

2) Lines 38: Zea mays should be italicized.

3) Lines 115-117: Reference should be added.

4) Line 118: Replace “bbelow” with “below”.

5) Line 134: A comma should be added after specifically.

6) Line 166: How did plants were uprooted?

7) Lines 168-169: How did rhizosphere soil was collected?

8) Lines 169-170: Arbuscular Mycorrhizal Fungi should be written in lowercase.

9) The equation for the total mycorrhization percentage should be mentioned.

10) Lines 179 and 192: The magnification unit should be standardized.

11) Reference should be added for pH measurement.

12) Reference should be added for exchangeable cations measurement.

13) The instruments and chemicals should be mentioned which were used to determine soil analysis.

14) Line 211: Extra full-stop should be deleted.

15) Lines 38: Full-stop should be deleted after algorithms.

16) In headings 2.7.1, 2.7.2, and 2.7.3, names and versions of software used for analysis should be mentioned.

17) The headings format should be uniform throughout the manuscript.

18) p should be italicized throughout the manuscript.

19) Line 278: Anova should be capitalized and followed throughout the manuscript.

20) Lines 285-287: It should be mentioned that these sites belong to which territory.

21) Line 289: Spore density unit should be corrected (100 g-1 soil) and followed throughout the manuscript.

22) Lines 365-366: Arbuscular Mycorrhizal Fungi should be written in lowercase and you have already mentioned the abbreviation earlier, so you have to use an abbreviation instead of using the full name. Follow it throughout the manuscript.

23) Lines 432-435: What are the favorable conditions that promote spore germination?

24) Line 443: Carbon (C) and phosphorus (P) abbreviations should be added at the start and no need to repeat it later.

25) References should be updated by adding journal names, italicizing the scientific names, and keeping the text in a similar pattern such as sentence form.

26) Figure 1 quality should be enhanced.

27) Figure 3, p should be italicized in the figure and caption, Anova should be capitalized, and p = < should be written in a standardized form and followed throughout the manuscript.

28) Figure 3, What do highland and lowland stand for? The author did not mention it in the whole manuscript.

29) Table 2, Zea mays L should be written as Zea mays L.

Reviewer #2: Dear,

The paper “Site climate more than soil properties and topography shape the natural arbuscular mycorrhizal symbiosis in maize and spore density within rainfed maize (Zea mays L.) cropland in the Eastern DR Congo” brings interesting data and relevant analyzes for understanding some edapho-climatic variables on some mycorrhizal parameters. Here are some observations about the text:

Summary

- It is well written, but it is important to add the study hypothesis.

Introduction

- Extremely long, with paragraphs with poorly connected construction;

- It is important to add the hypothesis at the end of the introduction text.

Methodology

- Inform whether the Trypan Blue, used to color the roots, was prepared in lactoglycerol or lactophenol;

- Inform the concentration of sucrose used to extract the glomerospores;

- It is important to build a heat map between the variables studied;

- Add algorithm analysis (K-means), to check which variables are most relevant in the study;

- It is important to add data on the Most Probable Number of AMF infective propagules, production of Soil Proteins Related to Glomalin (PSRG) and spore viability.

Discussion

- Needs adjustments, as there are excessive comparisons with other works. It is important to explain the observed behaviors, especially based on whether or not the hypothesis is confirmed;

- There is a need to further explore the effects of C and P on the mycorrhizal parameters evaluated;

- The discussion on sporulation needs to consider what is already well established for fungal reproduction, such as the buffering effect of the substrate on the production of glomerospores.

Conclusion

- They must be direct and written with the verb in the present tense; are written as a mix of results and final considerations.

Figures

- Figures 1 and 7 – improve resolution;

- Figures 2 and 3 – supplementary material.

Tables

- Delete table 3;

- Tables 5 and 6 – supplementary material

References

- It is important to add more up-to-date references on the topic.

After making the aforementioned adjustments, the paper can be considered for publication

Sincerely,

6. PLOS authors have the option to publish the peer review history of their article (what does this mean?). If published, this will include your full peer review and any attached files.

Reviewer #1: No

Reviewer #2: No

While revising your submission, please upload your figure files to the Preflight Analysis and Conversion Engine (PACE) digital diagnostic tool, https://pacev2.apexcovantage.com/. PACE helps ensure that figures meet PLOS requirements. To use PACE, you must first register as a user. Registration is free. Then, login and navigate to the UPLOAD tab, where you will find detailed instructions on how to use the tool. If you encounter any issues or have any questions when using PACE, please email PLOS at figures@plos.org. Please note that Supporting Information files do not need this step.</sites>

---

## [Author Response · Author response to Decision Letter 0]

6 Sep 2024

RESPONSES TO REVIEWERS

Dear Editors and Reviewers,

Please find attached the various responses to the comments raised during the review of my manuscript.

Thank you and best regards.

EDITOR COMMENTS 

--Figure 3: Mycorrhizal Status of maize (Zea mays L) in the three selected territory

--Replace by in the three selected territories or better, use

Response: The words “selected territory” was replaced by “selected territories”

REVIEWER #1: 

1) Lines 27: Units of spores density should be uniform.

Response : Spore density Unit was uniformed in all the manuscript

2) Lines 38: Zea mays should be italicized.

Response: the word “Zea mays” was italicized accordingly 

3) Lines 115-117: Reference should be added.

Responses : The reference was added accodingly

4) Line 118: Replace “bbelow” with “below”.

Response : the word was written correctly 

5) Line 134: A comma should be added after specifically.

Response : comma was added

6) Line 166: How did plants were uprooted?

Response: This is specified in the text; “The maize plants were delicately uprooted to preserve the integrity of the root system. A circle approximately 15 cm in diameter was traced around each plant, and the soil was dug to extract the entire root system”

7) Lines 168-169: How did rhizosphere soil was collected?

Responses: Collected roots, along with the surrounding rhizosphere soils (0-20-25 cm), were combined to create three composite samples

8) Lines 169-170: Arbuscular Mycorrhizal Fungi should be written in lowercase.

Response : Arbuscular Mycorrhizal Fungi was written accondingly

9) The equation for the total mycorrhization percentage should be mentioned.

Responses: The equation was added accordingly

10) Lines 179 and 192: The magnification unit should be standardized.

Responses: The magnification unit was standardized

11) Reference should be added for pH measurement.

Responses: Reference was added

12) Reference should be added for exchangeable cations measurement.

Responses: Reference was added

13) The instruments and chemicals should be mentioned which were used to determine soil analysis.

Response : some instrument and chemicals was mentioned in the text

14) Line 211: Extra full-stop should be deleted.

Response: Extra full-stop was deleted accondingly

15) Lines 38: Full-stop should be deleted after algorithms.

Response: Full-stop was deleted after algorithms accordingly 

16) In headings 2.7.1, 2.7.2, and 2.7.3, names and versions of software used for analysis should be mentioned.

Response: names and versions of software used for analysis was mentioned in the text

17) The headings format should be uniform throughout the manuscript.

Responses : The headings format was uniformized throughout the manuscript

18) p should be italicized throughout the manuscript.

Responses : p was italicized in the text 

19) Line 278: Anova should be capitalized and followed throughout the manuscript.

Responses : Anova was capitalized whirhin the manuscript

20) Lines 285-287: It should be mentioned that these sites belong to which territory.

Responses : The origine of each site was specified in the text 

21) Line 289: Spore density unit should be corrected (100 g-1 soil) and followed throughout the manuscript.

Responses : Spore density unit was corrected (100 g-1 soil) in all the manuscript

22) Lines 365-366: Arbuscular Mycorrhizal Fungi should be written in lowercase and you have already mentioned the abbreviation earlier, so you have to use an abbreviation instead of using the full name. Follow it throughout the manuscript.

Responses ; Arbuscular Mycorrhizal Fungi was written in lowercase

23) Lines 432-435: What are the favorable conditions that promote spore germination?

Responses : The favourable conditions for spore germination was specified in the text

24) Line 443: Carbon (C) and phosphorus (P) abbreviations should be added at the start and no need to repeat it later.

Responses : Carbon (C) and phosphorus (P) abbreviations was written correctly 

25) References should be updated by adding journal names, italicizing the scientific names, and keeping the text in a similar pattern such as sentence form.

Responses: All references cited was chacked and written correctely in the text and in the reference list

26) Figure 1 quality should be enhanced. 

Responses : Figure quality was improved. Se in the text (tables and figures file)

27) Figure 3, p should be italicized in the figure and caption, Anova should be capitalized, and p = < should be written in a standardized form and followed throughout the manuscript.

Response : This letters was generated with the R software as presented in the main text

28) Figure 3, What do highland and lowland stand for? The author did not mention it in the whole manuscript.

Response : highland and lowland stand for agroecological zones

29) Table 2, Zea mays L should be written as Zea mays L.

Response : the word was well written as Zea mays L.

REVIEWER #2:

Summary

- It is well written, but it is important to add the study hypothesis.

Response: The study hypothesis was added in the summary of the manuscript

Introduction

- Extremely long, with paragraphs with poorly connected construction; 

Response : The introduction was amended accordingly 

- It is important to add the hypothesis at the end of the introduction text.

Response :Hypothesis was added at the end of the introduction 

Methodology

- Inform whether the Trypan Blue, used to color the roots, was prepared in lactoglycerol or lactophenol;

Response : Trypan Blue, used to color the roots, was prepared in lactophenol. This were specified in the text

- Inform the concentration of sucrose used to extract the glomerospores;

Response : The sucrose concentrations used were specified in the text (two concentrations were used simultaneously during the centrifugation process: 20% sucrose and 60% sucrose).

- It is important to build a heat map between the variables studied;

Response : The heatmap between variable was added in the supplementary materialsl

- Add algorithm analysis (K-means), to check which variables are most relevant in the study;

Response : We used a multifactorial analysis (multiple factor analysis) that allowed us to group the sites based on their characteristics. We did not apply the K-means method.

- It is important to add data on the Most Probable Number of AMF infective propagules, production of Soil Proteins Related to Glomalin (PSRG) and spore viability.

Response : The Most Probable Number (MPN) was not evaluated during the study, nor was the dosage of Proteins Related to Glomalin (PSRG). However, for spore viability, it was specified that only viable spores were counted during the study

Discussion

- Needs adjustments, as there are excessive comparisons with other works. It is important to explain the observed behaviors, especially based on whether or not the hypothesis is confirmed;

Response : Some adjustments to the discussion have been made and the amendments have been added.

- There is a need to further explore the effects of C and P on the mycorrhizal parameters evaluated;

Response : We have amended the aspects regarding the effect of C and P on natural maize mycorrhization and spore density by presenting additional recent arguments.

- The discussion on sporulation needs to consider what is already well established for fungal reproduction, such as the buffering effect of the substrate on the production of glomerospores.

Responses : Too few new studies on AMF sporulation are available; however, we were able to add some recent information. See the article by Mukhongo et al., 2023.

Conclusion

- They must be direct and written with the verb in the present tense; are written as a mix of results and final considerations.

Response : The conclusion was amended accordingly

Figures

- Figures 1 and 7 – improve resolution;

- Figures 2 and 3 – supplementary material.

Response : Figure 1 and 7 resolution was improved, and Figure 2 was added in the supplementary material

Tables

- Delete table 3;

- Tables 5 and 6 – supplementary material

Response : Table 3 was deleted and Table 5 and 6 was puted in the supplementary material

References

- It is important to add more up-to-date references on the topic.

Response : some updated referencies was added in the text

---

## [Editor Report · Decision Letter 1]

10 Oct 2024

Site climate more than soil properties and topography shape the natural arbuscular mycorrhizal symbiosis in maize and spore density within rainfed maize (Zea mays L.) cropland in the Eastern DR Congo

PONE-D-24-15543R1

Dear Dr. Adrien Ndeko Byamungu,

We’re pleased to inform you that your manuscript has been judged scientifically suitable for publication and will be formally accepted for publication once it meets all outstanding technical requirements.

Kind regards,

Marcela Pagano, Ph.D, M.D.

Academic Editor

PLOS ONE

Additional Editor Comments (optional):

Manuscript improved accordingly to suggestions
---

## [Editor Report · Acceptance letter]

29 Oct 2024

PONE-D-24-15543R1 

PLOS ONE

Dear Dr. Ndeko, 

I'm pleased to inform you that your manuscript has been deemed suitable for publication in PLOS ONE. Congratulations! Your manuscript is now being handed over to our production team.

Kind regards, 

on behalf of

Dr. Marcela Pagano 

Academic Editor

PLOS ONE